# CoSy: Evaluating Textual Explanations of Neurons

**Laura Kopf**[1,2]    **Philine Lou Bommer**[1,3]    **Anna Hedström**[1,3,4]    **Sebastian Lapuschkin**[4]
**Marina M.-C. Höhne**[3,5]    **Kirill Bykov**[1,2,3]

[1]TU Berlin, Germany    [2]BIFOLD, Germany    [3]UMI Lab, ATB Potsdam, Germany
[4]Fraunhofer Heinrich-Hertz-Institute, Germany    [5]University of Potsdam, Germany

`kopf@tu-berlin.de`
`{pbommer,ahedstroem,mhoehne,kbykov}@atb-potsdam.de`
`sebastian.lapuschkin@hhi.fraunhofer.de`

## Abstract

A crucial aspect of understanding the complex nature of Deep Neural Networks (DNNs) is the ability to explain learned concepts within their latent representations. While methods exist to connect neurons to human-understandable textual descriptions, evaluating the quality of these explanations is challenging due to the lack of a unified quantitative approach. We introduce COSY (Concept Synthesis), a novel, architecture-agnostic framework for evaluating textual explanations of latent neurons. Given textual explanations, our proposed framework uses a generative model conditioned on textual input to create data points representing the explanations. By comparing the neuron's response to these generated data points and control data points, we can estimate the quality of the explanation. We validate our framework through sanity checks and benchmark various neuron description methods for Computer Vision tasks, revealing significant differences in quality. We provide an open-source implementation on GitHub[1].

## 1 Introduction

One of the key obstacles to the wider adoption of Machine Learning methods across various fields is the inherent opacity of modern Deep Neural Networks (DNNs)—in essence, we often lack an understanding of why these models make certain predictions. To address this problem, various explainability methods [1, 2] have been developed to make the decision-making processes of DNNs more understandable to humans. Explainability methods have broadened their focus from interpreting the decision-making of DNNs *locally*—for instance, interpreting specific inputs using saliency maps [3, 4, 5, 6]—to understanding the *global* behavior of models by analyzing individual model components and their functional purpose [7]. Following the latter global explainability approach, often referred to as *mechanistic interpretability* [8, 9, 10], some methods aim to describe the specific concepts neurons have learned to detect [11, 12, 13, 14, 15, 16], enabling analysis of how these high-level concepts influence network predictions.

A popular approach for explaining the functionality of latent representations of a network is to describe neurons using human-understandable textual concepts. A textual description is assigned to a neuron based on the concepts that the neuron has learned to detect or is significantly activated by. Over time, these methods have evolved from providing label-specific descriptions [11] to more complex compositional [12, 16] and open-vocabulary explanations [13, 15]. However, a significant challenge remains: the lack of a universally accepted quantitative evaluation measure for open-vocabulary neuron descriptions. As a consequence, different methods devised their own evaluation criteria, making it difficult to perform general-purpose, comprehensive cross-comparisons.

---

[1]`https://github.com/lkopf/cosy`

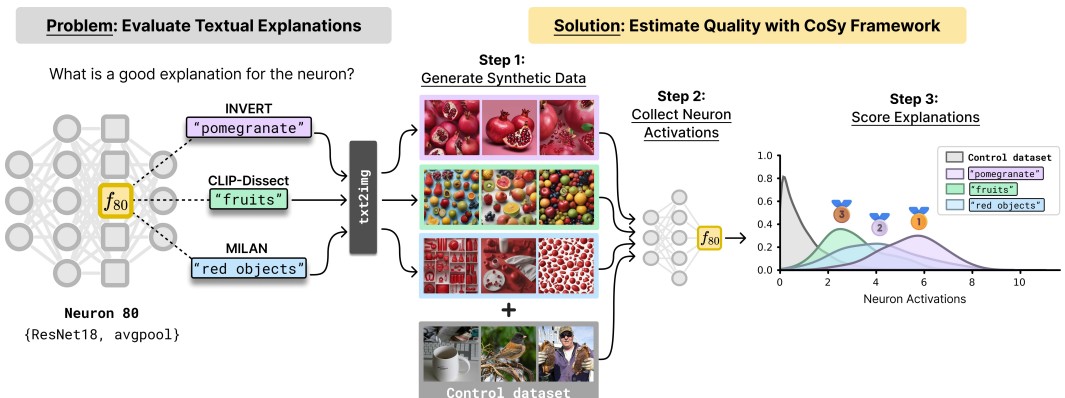

Figure 1: A schematic illustration of the CoSy evaluation framework for Neuron 80 in `ResNet18`'s avgpool layer. The current challenge lies in the absence of general-purpose, quantitative evaluation measures to benchmark textual explanations of neurons. To address this, we propose CoSy, a framework consisting of three steps: first, a generative model translates textual concepts into the visual domain, creating synthetic images for each explanation using a text-to-image model. Then, inference is performed on these synthetic images alongside a control image dataset to collect neuron activations. Finally, by comparing activations from the synthetic images with those from the control dataset, we quantitatively assess the quality of the textual explanation and compare results across different explanation methods. The implementation details of this example can be found in Appendix A.2.

In this work, we aim to bridge this gap by introducing Concept Synthesis (CoSy), the first automatic evaluation framework for textual explanations of neurons in Computer Vision (CV) models (illustrated in Figure 1). Our approach builds on recent advancements in Generative AI, which enable the generation of synthetic images that align with provided neuron explanations. We use a set of available text-to-image models to synthesize data points that are prototypical for specific target explanations. These data points allow us to evaluate how neurons differentiate between concept-related images and non-concept-related images combined in a control dataset. We summarize our contributions as below:

**(C1)** We provide the first general-purpose, quantitative evaluation framework CoSy (Section 3) that enables the evaluation of individual or a set of textual explanation methods for CV models.

**(C2)** In a series of sanity checks (Section 4), we analyze the choice of generative models and prompts for synthetic image generation, demonstrating framework reliability.

**(C3)** We benchmark existing explanation methods (Section 5) and extract novel insights, revealing substantial variability in the quality of explanations. Generally, textual explanations for lower layers are less accurate compared to those for higher layers.

## 2   Related Works

**Activation Maximization**   Activation Maximization is a commonly used methodology to understand what a neuron has learned to detect [17]. Such methods work by identifying input signals that trigger the highest activation in a neuron. This can be achieved synthetically, where an optimization process is employed to create the optimal input that maximizes the neuron's activation [18, 19, 20], or naturally, by finding such inputs within a data corpus [21]. Activation Maximization has been employed for explaining latent representations of models [22, 23], including probabilistic models [24], detection of backdoor attacks [25] and spurious correlations [26]. However, one of the key limitations of this methodology lies in its inability to scale, as it relies on users to manually audit maximization signals.

**Automatic Neuron Interpretation**   A more scalable alternative approach involves linking neurons with human-understandable concepts through textual descriptions. Network Dissection [11] (NetDissect) is a pioneering method in this field, associating convolutional neurons with a concept based on the Intersection over Union (IoU) of neuron activation maps and ground truth segmentation masks.

Table 1: Comparison of characteristics of neuron description methods. The columns (from left to right) represent the explanation method used, its textual output type (fixed-label, compositional, or open-vocabulary), the type of neuron targeted for analysis (convolutional, scalar, or predetermined), the target metric the method optimizes (IoU, WPMI, AUC, etc.), whether the method relies on auxiliary black-box models for finding or generating explanations (img2txt model, CLIP), and whether the explanation method is architecture-agnostic, meaning it can be applied to any CV model. For a more detailed description of each method, refer to Appendix A.1.

| Method | Explanation | Neuron Type | Target | Black-Box Dependency | Architecture-Agnostic |
|---|---|---|---|---|---|
| NetDissect [11] | fixed-label | conv. | IoU | — | ✓ |
| CompExp [12] | compositional | conv. | IoU | — | ✓ |
| MILAN [13] | open-vocabulary | conv. | WPMI | img2txt model | ✓ |
| FALCON [14] | open-vocabulary | predetermined | avg. CLIP score | CLIP | — |
| CLIP-Dissect [15] | open-vocabulary | scalar | SoftWPMI | CLIP | ✓ |
| INVERT [16] | compositional | scalar | AUC | — | ✓ |

Building on this, Compositional Explanations of Neurons (CompExp) [12] enhanced explanation detail by enabling the use of compositional concepts—i.e., concepts constructed with logical operators. MILAN [13] further expanded this by allowing for open-vocabulary explanations, permitting the generation of descriptions beyond predefined labels. INVERT [16] adopted a compositional concept approach, enabling explanations for general neuron types without the need for segmentation masks. It assigns compositional labels based on a neuron's ability to distinguish concepts, using the Area Under the Receiver Operating Characteristic Curve (AUC). FALCON [14] and CLIP-Dissect [15] compute image-text similarity with a CLIP model [27] for the most activating images and their corresponding captions or concept sets. Each method defines its optimization criteria, lacking a unified consensus on what constitutes a good explanation. For detailed descriptions of the methods and their optimization objectives, please refer to Appendix A.1. An overview of the different techniques is illustrated in Table 1.

**Prior Methods for Evaluation**   While significant effort has been made towards developing approaches and tools for evaluating *local* explanations [28, 29, 30], there has been relatively limited focus on evaluating *global* methods, in particular neuron description methods. Currently, to the best of our knowledge, there is no unified approach that allows for benchmarking across models and explanation methods. In their respective papers, the INVERT and CLIP-Dissect explanation methods evaluated the accuracy of their explanations by comparing the generated neuron labels with ground truth descriptions provided for neurons in the output layer of a network. However, this evaluation is limited to output neurons and fixed labels only. CLIP-Dissect additionally evaluates the quality of explanations by computing the Cosine Similarity in a sentence embedding space between the ground truth class name for each neuron and the explanation generated by the method. FALCON employs a human study conducted on Amazon Mechanical Turk to evaluate the concepts generated by the method. Participants are tasked with selecting the best explanation for each target feature from a selection of explanation methods, considering a given set of highly and lowly activating images. MILAN evaluates the performance of neuron labeling methods relative to human annotations using BERTScores [31]. While human studies are generally beneficial, the conventional setup can be misleading and may fail to fully capture the intended evaluation criteria, introducing potential biases. Typically, annotators describe the images that most strongly activate a neuron, and these descriptions are then compared to an automatic explanation. However, this approach primarily evaluates the alignment with the most activating images rather than the accuracy of the explanation in describing the neuron's function. Moreover, these highly activating images may not accurately represent the neuron's overall behavior, as they only reflect the maximum tail of the distribution.

## 3   Method

In the following section, we introduce COSY—a first automatic evaluation procedure for open-vocabulary textual explanations for neurons. We first define preliminary notations in Section 3.1, then describe COSY formally in Section 3.2.

### 3.1 Preliminaries

Consider a Deep Neural Network (DNN) represented by the function $g : \mathcal{X} \to \mathcal{Z}$, where $\mathcal{X} \subset \mathbb{R}^{h \times w \times c}$ denotes the input image domain and $\mathcal{Z} \subset \mathbb{R}^l$ represents the model's output domain. We can view the model as a composition of two functions, $F : \mathcal{X} \to \mathcal{Y}$, and $L : \mathcal{Y} \to \mathcal{Z}$, such that $g = L \circ F$. Here $\mathcal{Y} \subset \mathbb{R}^{d \times w^* \times h^*}$, where $d \in \mathbb{N}$ is the number of neurons in the layer, and $w^*, h^* \in \mathbb{N}$ represent the width and height of the feature map, respectively. The function $F$, which we refer to as the *feature extractor*, can be chosen based on the layer of the model we aim to inspect. This could be an existing layer within the model or a concept bottleneck layer [32]. We refer to the $i$-th neuron within the layer as $f_i(\boldsymbol{x}) = F_i(\boldsymbol{x}) : \mathcal{X} \to \mathbb{R}^{w^* \times h^*}$. Within the scope of this paper, we refer to *explanation method* as an operator $\mathcal{E}$ that maps a neuron to the textual description $s = \mathcal{E}(f_i) \in \mathcal{S}$, where $\mathcal{S}$ is a set of potential textual explanations. The specific set of explanations depends on the implementation of the particular method (see Appendix A.1).

### 3.2 CoSy: Evaluating Open-Vocabulary Explanations

We assume that a good textual explanation for a neuron should provide a human-understandable description of an input that strongly activates the neuron. However, modern methods for explaining the functional purpose of neurons often provide open-vocabulary textual explanations, complicating the quantitative collection of natural data that represents the explanation. To address this issue, CoSy utilizes recent advancements in generative models to synthesize data points that correspond to the textual explanation. The response of a neuron to a set of synthetic images is measured and compared to the neuron's activation on a set of control natural images representing random concepts. This comparison allows for a quantitative evaluation of the alignment between the explanation and the target neuron.

The parameters of the proposed method include a control dataset $\mathbb{X}_0 = \{\boldsymbol{x}_1^0, \ldots, \boldsymbol{x}_n^0\} \subset \mathcal{X}, n \in \mathbb{N}$, which consists of natural images representing the concepts on which the model was originally trained. Additionally, it incorporates a generative model $p_M$ used for synthesizing images, along with a specified number of generated images $m \in \mathbb{N}$. The control dataset typically includes a balanced selection of validation class images. Given a neuron $f_i$ and explanation $s \in \mathcal{S}$, CoSy evaluates the alignment between the explanation and a neuron in three consecutive steps, which are illustrated in Figure 1.

1. **Generate Synthetic Data.** The first step involves generating synthetic images for a given explanation $s \in \mathcal{S}$, which we use as a prompt to a generative model $p_M$ to create a collection of synthetic images, denoted as $\mathbb{X}_1 = \{\boldsymbol{x}_1^1, \ldots, \boldsymbol{x}_m^1\} \sim p_M(\boldsymbol{x} \mid s)$. This collection consists of $m \in \mathbb{N}$ images, where $m$ is adjustable as a parameter of the evaluation procedure.

2. **Collect Neuron Activations.** Given the control dataset $\mathbb{X}_0$ and the set of generated synthetic images $\mathbb{X}_1$, we collect activations as follows:

$$
\begin{aligned}
\mathbb{A}_0 &= \{\sigma(f_i(\boldsymbol{x}_1^0)), \ldots, \sigma(f_i(\boldsymbol{x}_n^0))\} \in \mathbb{R}^n, \\
\mathbb{A}_1 &= \{\sigma(f_i(\boldsymbol{x}_1^1)), \ldots, \sigma(f_i(\boldsymbol{x}_m^1))\} \in \mathbb{R}^m,
\end{aligned}
\tag{1}
$$

where $\sigma : \mathbb{R}^{w^* \times h^*} \to \mathbb{R}$ is an aggregation function for multi-dimensional neurons. Within the scope of our paper, we use Average Pooling as aggregation function

$$
\sigma(\boldsymbol{y}) = \frac{1}{w^* h^*} \sum_{k \in [1, w^*], l \in [1, h^*]} \boldsymbol{y}_{k,l}, \quad \boldsymbol{y} \in \mathcal{Y} \subset \mathbb{R}^{w^* \times h^*}.
\tag{2}
$$

3. **Score Explanations.** The final step of the proposed method relies on the evaluation of the difference between neuron activations on the control dataset $\mathbb{A}_0$ and neuron activations given the synthetic dataset $\mathbb{A}_1$. To quantify this difference, we utilize a *scoring function* $\Psi : \mathbb{R}^n \times \mathbb{R}^m \to \mathbb{R}$ to measure the difference between the distributions of activations.

In the context of our paper, we employ the following scoring functions:

- **Area Under the Receiver Operating Characteristic (AUC)**
  AUC is a widely used non-parametric evaluation measure for assessing the performance of binary classification. In our method, AUC measures the neuron's ability to distinguish

between synthetic and control data points

$$\Psi_{\text{AUC}}(\mathbb{A}_0, \mathbb{A}_1) = \frac{\sum_{a \in \mathbb{A}_0} \sum_{b \in \mathbb{A}_1} \mathbf{1}[a < b]}{|\mathbb{A}_0| \cdot |\mathbb{A}_1|}. \tag{3}$$

- **Mean Activation Difference (MAD)**

  MAD is a parametric measure that quantifies the difference between the mean activation of the neuron on synthetic images and the mean activation on control data points

  $$\Psi_{\text{MAD}}(\mathbb{A}_0, \mathbb{A}_1) = \frac{\frac{1}{m} \sum_{b \in \mathbb{A}_1} b - \frac{1}{n} \sum_{a \in \mathbb{A}_0} a}{\sqrt{\frac{1}{n-1} \sum_{a \in \mathbb{A}_0} (a - \bar{a})^2}}, \tag{4}$$

  with mean control activation $\bar{a} = \frac{1}{n} \sum_{a \in \mathbb{A}_0} a$.

These two chosen metrics complement each other. AUC, being non-parametric and stable to outliers, evaluates the classifier's ability to rank synthetic images higher than control images (with scores ranging from 0 to 1, where 1 represents a perfect classifier and 0.5 is random). On the other hand, MAD allows us to parametrically measure the extent to which images corresponding to explanations maximize neuron activation.

## 4 Sanity Checks

To ensure the reliability of our proposed evaluation measure, all steps within our framework need to be subject to sanity checks [33]. In this section, we analyze the following: (1) which generative models and prompts provide the best similarity to natural images, (2) whether the model's behavior on synthetic and natural images differs for the same class, and (3) validating that COSY provides appropriate evaluation scores for true and random explanations, given a known ground truth class for the neuron.

### 4.1 Synthetic Image Reliability

One of the key features of COSY is its reliance on generative models to translate textual explanations of neurons into the visual domain. Thus, it is essential that the generated images reliably resemble the textual concepts. In the following section, we present an experiment where we varied several parameters of the generation procedure and evaluated the visual similarity between generated images and natural ones, focusing on concepts for which we have a collection of natural images.

For our analysis, we used only open-source and freely available text-to-image models, namely Stable Diffusion XL 1.0-base (SDXL) [34] and Stable Cascade (SC) [35]. We also varied the prompts for image generation. To measure the similarity between synthetic images and natural images corresponding to the same concept, we used Cosine Similarity (*CS*) in the CLIP embedding space with the CLIP-ViT-B/32 model [27]. We select a set of 50 random concepts from the 1,000 classes in the ImageNet validation dataset [36]. For each `[concept]` we use five different prompts and employ them with SDXL and SC models, generating 50 images per class. We then measure the *CS* between image pairs of the same class.

Figure 2 illustrates the comparison across all generative models and prompts in terms of *CS* of generated images to natural images of the same class. The results indicate that when using Prompt 5 as input to SDXL, the synthetic images show the highest similarity to natural images. The performance is generally best with the most detailed prompt (5) and closely aligns with prompts 1, 3, and 4. Moreover, SDXL appears to be slightly more effectively realizing detailed prompts than SC. As anticipated, the poorly constructed prompt (2) results in the lowest similarity to natural images for both models. To address prompt bias and dataset dependency, we compare the object-focused ImageNet with the scene-focused Places365 (see Appendix A.3). We find that close-up prompts work well for object-centric datasets, while general prompts like "photo of" are better for scene-based datasets. If not stated otherwise, for all following experiments, Prompt 5 together with SDXL model was employed for image generation.

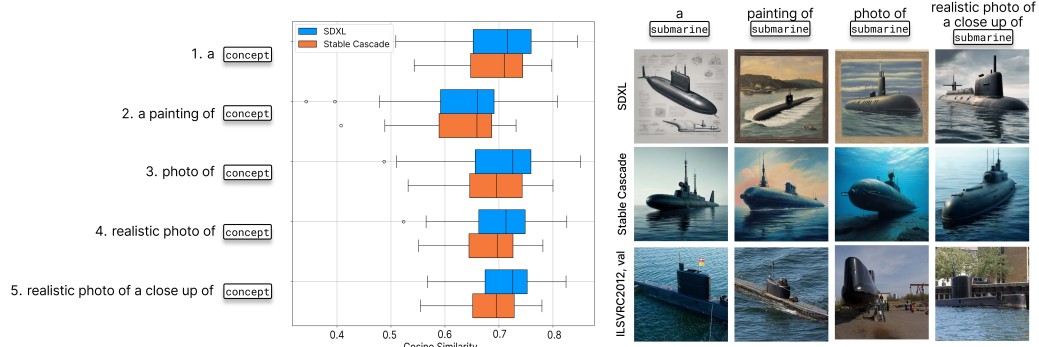

Figure 2: An overview of the impact of varying the prompt on the similarity between natural and synthetic images, using two text-to-image models. Left: average Cosine Similarity (*CS*) across all natural and synthetic images over all classes are reported. Higher *CS* values are better, indicating greater similarity between the images. Right: an illustration of the visual differences produced by the SDXL and SC models in response to diverse prompts for the concept "submarine", and natural images from the ImageNet validation dataset [36]. Our results show that both SDXL and SC generate similar images, with SDXL generally being more closely aligned with natural images than SC.

## 4.2    Do Models Respond Differently to Synthetic and Natural Images?

Given the visual similarity between natural and synthetic images of the same class, we investigate whether CV models respond differently to these groups and if the activation differences indicate adversarial behavior. To this end, we employed four different models pre-trained on ImageNet: ResNet18 [37], DenseNet161 [38], GoogLeNet [39], and ViT-B/16 [40]. For each model, we randomly selected 50 output classes and generated 50 images per class using the class descriptions. We pass both synthetic and natural images through the models, collecting the activations of the output neuron corresponding to each class.

Figure 3 (a) illustrates the distributions of the MAD between synthetic and natural images for the same class across the 50 classes. Across all models, we observe that the median activation of synthetic images is slightly higher than that of natural images of the same class. However, this difference is small, given the 0 value lies within 1 standard deviation. We also illustrate the activations of neuron $504$ in the ResNet18 output layer for the "coffee mug" class in Figure 3 (b). The results indicate a strong overlap in the neural response to both synthetic and natural images. While synthetic images activate the neuron slightly more, this does not constitute an artifactual behavior or affect our framework, which we demonstrate in the following experiment.

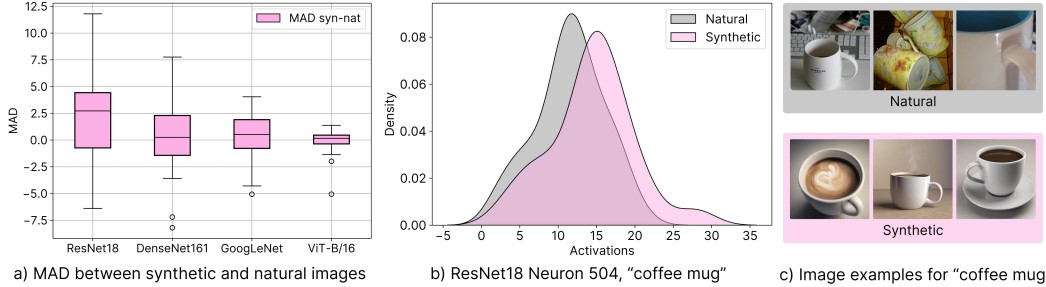

a) MAD between synthetic and natural images    b) ResNet18 Neuron 504, "coffee mug"    c) Image examples for "coffee mug"

Figure 3: An overview of analyses performed to study the similarity between natural and synthetic images. From left to right: (a) an overview of MAD scores between synthetic and natural image activations of the output neuron's ground truth classes for each model studied in this work, (b) activations collected for neuron $504$ in ResNet18 for the class "coffee mug", showcasing the difference between the natural and synthetic distributions and (c) examples of natural versus synthetic images. In both analyses, we observe a substantial overlap in the activations of synthetic and natural images, suggesting that the models respond similarly to both types of images.

Table 2: Comparison of true and random explanations on output neurons with known ground truth labels. This table presents the average quality scores (with standard deviations) for true explanations, derived from target class labels, and random explanations, derived from randomly selected synthetic image classes (including the target class), across four models pre-trained on ImageNet. Higher values are better. Our results consistently show high scores for true explanations and low scores for random ones.

| Model | AUC (↑) | | MAD (↑) | |
|---|---|---|---|---|
| | True | Random | True | Random |
| ResNet18 | 0.98±0.09 | 0.47±0.21 | 6.46±2.07 | -0.11±0.79 |
| DenseNet161 | 0.99±0.08 | 0.44±0.22 | 7.11±1.82 | -0.19±0.73 |
| GoogLeNet | 0.99±0.07 | 0.48±0.23 | 7.74±2.14 | -0.06±0.78 |
| ViT-B/16 | 0.99±0.05 | 0.49±0.22 | 13.12±3.19 | 0.09±1.10 |

## 4.3 Random Baseline

A robust evaluation metric should reliably discern between random explanations resulting in low scores and non-random explanations resulting in high scores. To assess our evaluation framework regarding this requirement, we evaluated the results of the CoSY evaluation by comparing the scores of ground truth explanations with those of randomly selected explanations.

Following the experimental setup in Section 4.2, we selected a set of 50 output neurons and compared the CoSY scores of the ground truth explanations, given by the neuron label, with those of randomly selected explanations. The results, presented in Table 2, consistently demonstrate high scores for true explanations and low scores for random explanations. This experiment provides further evidence supporting the correctness of the proposed evaluation procedure. An additional experiment that excludes the target class from the control dataset is presented in Appendix A.4, along with an analysis of the robustness of the evaluation measure detailed in Appendix A.6.

## 5 Evaluating Explanation Methods

Within the scope of this section, we produce a comprehensive cross-comparison of various methods for the textual explanations of neurons. For this comparison, we employed models trained on different datasets, and we conducted our analysis on the latent layers of the models, where no ground truth is known.

### 5.1 Benchmarking Explanation Methods

In this section, we evaluated three recent textual explanation methods, namely MILAN, INVERT, and CLIP-Dissect. Our analysis involves six distinct models: four pre-trained on the ImageNet dataset [36] (ResNet18 [37], ResNet50 [37], ViT-B/16 [40], DINO ViT-S/8 [41]) and two pre-trained on the Places365 dataset [42] (DenseNet161 [38], ResNet50 [37]). The ImageNet dataset focuses on objects, whereas the Places365 dataset is designed for scene recognition. Consequently, we customized our prompts accordingly: Prompt 5 performs best for object recognition, while for scene recognition, we found that Prompt 3 is more effective. Therefore, Prompt 3 was utilized in the Places365 experiment.

For generating explanations with the explanation methods, we use a subset of 50,000 images from the training dataset on which the models were trained. For evaluation with CoSY, we use the corresponding validation datasets the models were pre-trained on as the control dataset. Additionally, for CLIP-Dissect, we define concept labels by combining the 20,000 most common English words with the corresponding dataset labels. For INVERT we set the compositional length of the explanation as $L = 1$, where $L \in \mathbb{N}$. For more details on compute resources, refer to Appendix A.7.

Results of the evaluation can be found in Table 3. Overall, INVERT achieves the highest AUC scores across all models and datasets, except for DINO ViT-S/8 and ResNet18 applied to ImageNet, where CLIP-Dissect achieves a higher or similar score. Also across other models and datasets, CLIP-Dissect demonstrates consistently good results. Since INVERT optimizes AUC in explanation generation, it

Table 3: Benchmarking of neuron description methods, for neurons in the second to last layer across different models. Explanations are generated for a randomly selected set of 50 neurons, with average scores for both AUC and MAD reported alongside standard deviations. Higher values indicate better performance; **bold** numbers represent the highest scores.

| Dataset | Model | Layer | Method | AUC (↑) | MAD (↑) |
|---------|-------|-------|--------|---------|---------|
| ImageNet | ResNet18 | Avgpool | MILAN | 0.61±0.23 | 0.69±1.35 |
| | | | CLIP-Dissect | **0.93±0.11** | **3.85±1.88** |
| | | | INVERT | **0.93±0.11** | 3.23±1.72 |
| | ResNet50 | Avgpool | MILAN | 0.44±0.23 | -0.08±0.72 |
| | | | CLIP-Dissect | 0.95±0.08 | **4.98±2.57** |
| | | | INVERT | **0.96±0.06** | 4.62±2.26 |
| | ViT-B/16 | Features | MILAN | 0.53±0.19 | 0.12±0.76 |
| | | | CLIP-Dissect | 0.78±0.19 | 1.29±1.01 |
| | | | INVERT | **0.89±0.17** | **1.67±0.82** |
| | DINO ViT-S/8 | Layer 11 | MILAN | 0.59±0.21 | 0.37±0.91 |
| | | | CLIP-Dissect | **0.95±0.08** | **4.59±2.62** |
| | | | INVERT | 0.73±0.27 | 2.70±3.48 |
| Places365 | DenseNet161 | Features | MILAN | 0.56±0.28 | 0.44±1.30 |
| | | | CLIP-Dissect | 0.82±0.21 | **2.52±2.33** |
| | | | INVERT | **0.85±0.16** | 2.21±1.95 |
| | ResNet50 | Avgpool | MILAN | 0.65±0.28 | 1.11±1.67 |
| | | | CLIP-Dissect | 0.92±0.11 | **3.73±2.39** |
| | | | INVERT | **0.94±0.08** | 3.54±1.99 |

may be biased towards AUC in our evaluation, leading to higher scores. MILAN generally performs poorly, with an average AUC below 0.65 across all tasks, indicating performance close to random guessing. MILAN tends to generate highly abstract explanations, such as "white areas", "nothing" or "similar patterns". These abstract concepts are particularly challenging for a text-to-image model to generate accurately, likely contributing significantly to the low scores of MILAN. Contrary to the AUC scores, the MAD scores suggest that CLIP-Dissect outperforms INVERT for convolutional neural networks applied to both datasets. Nonetheless, in these cases, INVERT concepts also achieve consistently high scores. Otherwise, we find similar outcomes for both metrics $\Psi$, with MILAN achieving poor scores in all experimental settings.

## 5.2 Explanation Methods Struggle to Explain Lower Layer Neurons

In addition to the general benchmarking, we aimed to study the quality of explanations for neurons in different layers of a model. Since it is well known that lower-layer neurons usually encode lower-level concepts [43], it is interesting to see whether explanation methods can capture the concepts these neurons detect. To investigate this, we examined the quality of explanations across layers 1 to 4 and the output layer of an ImageNet pre-trained `ResNet18`. In addition to three prior explanation methods, we included the FALCON method in our analysis. For more details on the implementation of FALCON see Appendix A.1.4, for additional results of the original FALCON implementation see Appendix A.8, and for qualitative examples and a discussion of lower-level concepts see Appendix A.9. For each layer, we randomly selected 50 neurons for analysis.

In Figure 4 we present the AUC and MAD results for all explanation methods across layers 1 to 4 and the output layer of `ResNet18`. While less pronounced for the AUC metric, in general, we find increasing scores for later layers across all methods and both metrics $\Psi$, which suggest higher concept quality in later layers. Furthermore, we find that similar to the benchmarking experiments, MILAN achieves lower scores across metrics. Both MILAN and FALCON consistently show lower performance, with AUC scores of $0.5$ indicating random guessing. Nonetheless, we point out that these methods typically output semantically high-level concepts. Potentially, this is related to the inherent difficulty in describing low-level abstractions in natural language given their complexity (see Appendix A.10).

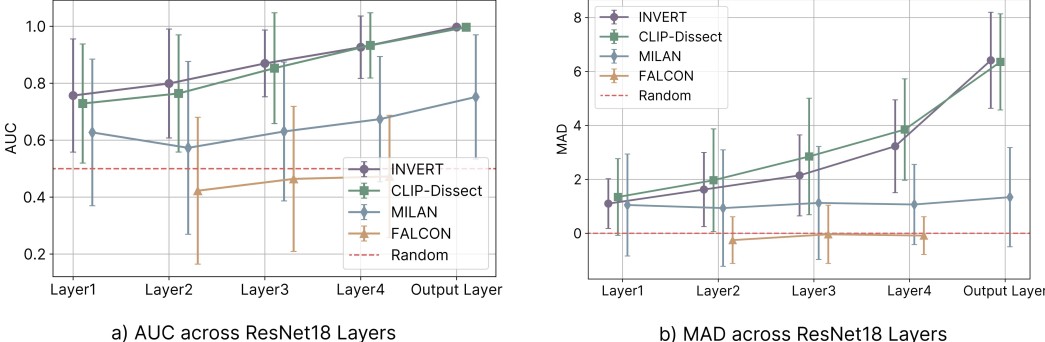

a) AUC across ResNet18 Layers

b) MAD across ResNet18 Layers

Figure 4: A comparison of how different explanation methods vary in their quality, as measured by (a) AUC and (b) MAD, across different layers in `ResNet18`. INVERT and CLIP-Dissect maintain high AUC and MAD scores across all layers, while MILAN and FALCON have lower scores. Overall, performance declines in the lower layers for all methods.

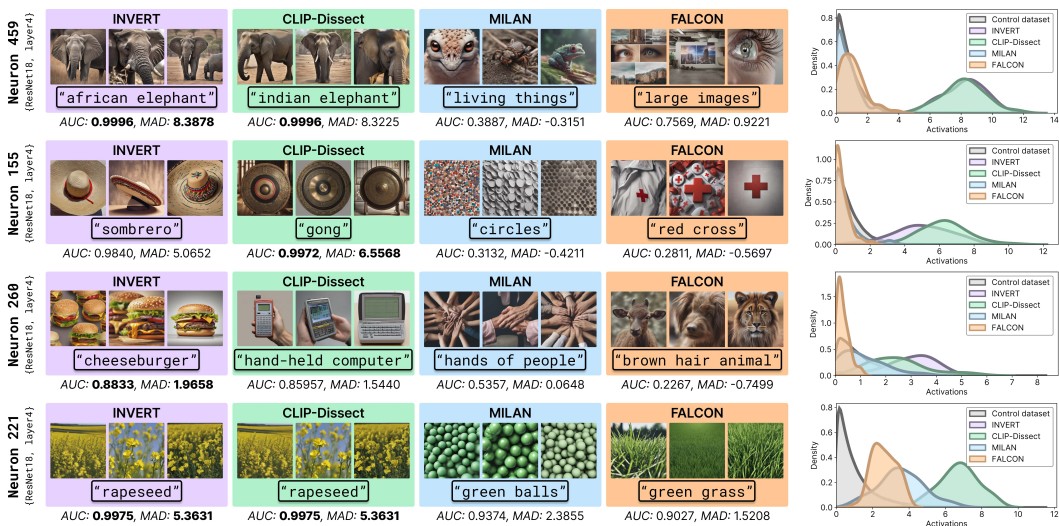

Figure 5: A qualitative example, of neuron explanations across four neurons. The first four panels include the textual explanation across INVERT, FALCON, CLIP-Dissect, and MILAN alongside three corresponding generated images. The respective AUC and MAD scores are reported below each panel. The last panel shows the activation distributions across 50 generated images for each method and the distribution of the control data.

## 5.3 What are Good Explanations?

In our approach, we propose that testing visual representations of textual explanations on neurons can provide insights into what constitutes good explanations. Building on this premise, we observe consistently high results from CLIP-Dissect and INVERT. The qualitative examples in Figure 5 demonstrate that their explanations share visually similar concepts (neurons 155 and 459) or even identical concepts (neuron 221) while both achieving high AUC and MAD scores. It is important to note that although INVERT performs slightly better in several tasks, the explanations are constrained to the input data labels. In contrast, CLIP-Dissect can generate descriptions from a broader selection of concepts, though its reliance on a black-box model reduces interpretability compared to INVERT.

There are instances, such as neuron 260 in Figure 5, where all explanations vary significantly. In these cases, we find that the explanation activation distributions of FALCON and MILAN often overlap with or even match the control dataset, providing the user with nearly random explanations. This observation aligns with our overall findings: both the AUC and MAD scores consistently reflect the

low performance of FALCON and MILAN explanations in the COSY evaluation. Also, neurons $459$ and $155$ demonstrate the gap between consistently higher and lower-performing explanation methods.

# 6   Conclusion

In this work, we propose the first automatic evaluation framework for textual explanations of neurons. Unlike existing ad-hoc evaluation methods, we can now quantitatively compare different neuron description methods against each other and test, whether the given explanation describes the neuron accurately, based on its activations. We can evaluate the quality of individual neuron explanations by examining how accurately they align with the generated concept data points, without requiring human involvement.

Our comprehensive sanity checks demonstrate that COSY guarantees a reliable explanation evaluation. In several experiments, we show that neuron description methods are most applicable for the last layers, where high-level concepts are learned. In these layers, INVERT and CLIP-Dissect provide high-quality neuron concepts, whereas MILAN and FALCON explanations have lower quality and can present close to random concepts, which might lead to wrong conclusions about the network. Thus, the results highlight the importance of evaluation when using neuron description methods.

**Limitations**   The use of generative models involves a distinct set of limitations. For instance, text-to-image models may not include certain concepts in their training data, which can reduce their generative performance. This limitation, however, can often be addressed by analyzing the pre-training datasets and assessing model performance. Moreover, the model's capabilities of generating highly abstract concepts like "white objects" can be limited. However, the challenges with abstract concepts also reflect the descriptive quality of the provided explanations—explanations should be inherently understandable to humans. In both cases, exploring more sophisticated, specialized, or constrained models may offer improvement.

**Future Work**   Evaluation of non-local explanation methods is still a largely neglected research area, where COSY plays an important yet preliminary part. In the future, we need additional, complementary definitions of explanation quality that extend our precise definition of AUC and MAD, e.g., that involve humans to assess plausibility [44] or evaluate explanation quality via the success of a downstream task [45]. Furthermore, we plan to extend the application of our evaluation framework to additional domains including NLP and healthcare. In particular, it would be interesting to analyze the quality of more recent autointerpretable explanation methods given by highly opaque, large language models (LLMs) [46, 9]. We believe that applying COSY to healthcare datasets, where high-quality explanations are crucial, represents an impactful next step.

# Acknowledgements

This work was partly funded by the German Ministry for Education and Research (BMBF) through the project Explaining 4.0 (ref. 01IS200551). Additionally, this work was supported by the European Union's Horizon Europe research and innovation programme (EU Horizon Europe) as grant TEMA (101093003); the European Union's Horizon 2020 research and innovation programme (EU Horizon 2020) as grant iToBoS (965221); and the state of Berlin within the innovation support programme ProFIT (IBB) as grant BerDiBa (10174498).

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

# A   Appendix

## A.1   Neuron Description Methods

Neuron description methods aim to provide insights into human-understandable concepts learned by DNNs, enabling a deeper understanding of their decision-making mechanisms. These methods provide textual descriptions for neurons in CV models. This creates a connection between the abstract representation of a concept by the neural network and a human interpretation. In general, a concept can be any abstraction, such as a color, an object, or even an idea [47]. Textual explanations of a neuron $f_i$ can originate from various spaces depending on their generation process.

As defined in Section 3.1, we refer to *explanation method* as an operator $\mathcal{E}$ that maps a neuron to the textual description $s = \mathcal{E}(f_i) \in \mathcal{S}$, where $\mathcal{S}$ is a set of potential textual explanations. The specific set of explanations depends on the implementation of the particular method. We define the following subsets of textual descriptions $s \in \mathcal{S}$:

- $\mathcal{C}$ represents the space of individual concepts,
- $\mathcal{L}$ represents the space of logical combinations of concepts,
- $\mathcal{N}$ represents the space of open-ended natural language concept descriptions.

These textual descriptions serve as explanations for $f_i$ generated by explanation methods.

Examples for such explanation methods are MILAN [13], FALCON [14], CLIP-Dissect [15], and INVERT [16]. Figure 6 shows the general principle of how $\mathcal{E}$ works. In Table 4 we outline the origin of textual descriptions and their corresponding set memberships for each $\mathcal{E}$.

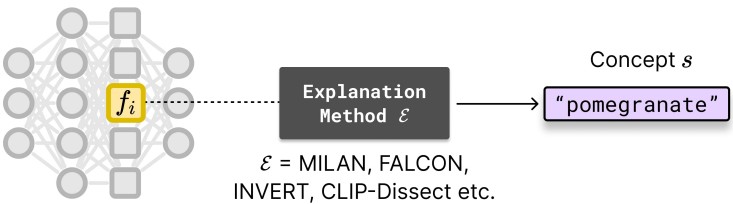

Figure 6: Neuron Description Methods. A neuron $f_i$ is selected, and a neuron description method $\mathcal{E}$ is applied to generate a textual description $s$ explaining $f_i$.

Table 4: Set Membership and Origin of Descriptions. Generated textual descriptions $s$ have varying set membership and origin across all $\mathcal{E}$. These descriptions can originate from distinct spaces: individual concepts $\mathcal{C}$, logical combinations of concepts $\mathcal{L}$, and open-ended natural language concept descriptions $\mathcal{N}$. A labeled dataset refers to a collection of images paired with individual concept labels. Generated captions are produced by image-to-text models, such as Show-Attend-Tell [48]. An image caption dataset consists of image-caption pairs. A concept set consists of textual concept labels.

| Method | Set | Origin |
|---|---|---|
| NetDissect | $\mathcal{C}$ | labeled dataset |
| CompExp | $\mathcal{L}$ | labeled dataset |
| MILAN | $\mathcal{N}$ | generated caption |
| FALCON | $\mathcal{N}$ | image caption dataset |
| CLIP-Dissect | $\mathcal{C}$ | concept set |
| INVERT | $\mathcal{L}$ | labeled dataset |

### A.1.1   NetDissect

Network Dissection (NetDissect) [11] is a method designed to explain individual neurons of DNNs, particularly convolutional neural networks (CNNs) within the domain of CV. This approach systematically analyzes the network's learned concepts by aligning individual neurons with given semantic

concepts. To perform this analysis, annotated datasets with segmentation masks are required, where these masks label each pixel in an image with its corresponding object or attribute identity. The Broadly and Densely Labeled Dataset (Broden) [11] combines a set of densely labeled image datasets that represent both low-level concepts, such as colors, and higher-level concepts, such as objects. It provides a comprehensive set of ground truth examples for a broad range of visual concepts such as objects, scenes, object parts, textures, and materials in a variety of contexts.

A concept $s \in \mathcal{C} \subset \mathcal{S}$ is defined as a visual concept in NetDissect and is provided by the pixel-level annotated Broden dataset. Given a CNN and the Broden dataset as input, NetDissect explains a neuron $f_i$ by searching for the highest similarity between concept image segmentation masks and neuron activation masks. Concept image segmentation masks are provided by the Broden dataset $B_s(\boldsymbol{x}) \in \{0,1\}^{H \times W}$, where a value of 1 signifies the pixel-level presence of $s$, and 0 denotes its absence. Neuron activation masks are obtained by thresholding the continuous neuron activations of $f_i$ into binary masks $A(\boldsymbol{x}) \in \{0,1\}^{H \times W}$. Then the similarity $\delta_{\text{IoU}}$ between image segmentation masks and binary neuron masks can be evaluated using the Intersection over Union score (IoU) for an individual neuron within a layer:

$$\delta_{\text{IoU}}(f_i, s) = \frac{\sum_{\boldsymbol{x} \in \boldsymbol{X}} \mathbf{1}\left(B_s(\boldsymbol{x}) \cap A(\boldsymbol{x})\right)}{\sum_{\boldsymbol{x} \in \boldsymbol{X}} \mathbf{1}\left(B_s(\boldsymbol{x}) \cup A(\boldsymbol{x})\right)}. \tag{5}$$

The NetDissect method is optimized to identify the concept that yields the highest IoU score between binary masks and image segmentation masks. This can be formalized as:

$$\mathcal{E}_{\text{NetDissect}}(f_i) = \arg\max_{s \in \mathcal{C} \subset \mathcal{S}} \delta_{\text{IoU}}\left(f_i, s\right). \tag{6}$$

NetDissect is constrained to segmentation datasets, relying on pixel-level annotated images with segmentation masks. Moreover, its labeling capabilities are confined to concepts provided within a labeled dataset. Furthermore, only individual concepts can be associated with each neuron.

### A.1.2 CompExp

To overcome the limitation of explaining neurons with only a single concept, the Compositional Explanations of Neurons (CompExp) method was later introduced [12], enabling the labeling of neurons with compositional concepts. The method obtains its explanations by merging individual concepts into logical formulas using composition operators AND, OR, and NOT. The formula length $L \in \mathbb{N}$ is defined beforehand. The initial stage of explanation generation is similar to NetDissect, a set of images is taken as input, and convolutional neuron activations are converted into binary masks. The explanations are constructed through a beam search algorithm [49], beginning with individual concepts and gradually building them into more complex logical formulas. Throughout the beam search stages, the existing formulas in the beam are combined with new concepts. These new formulas are measured by the IoU. The maximization of the IoU score is desired to get a high explanation quality.

The approach for obtaining $\delta_{\text{IoU}}$ is the same as in Equation 5. In contrast to NetDissect, the explanations can be a combination of concepts, where $s \in \mathcal{L} \subset \mathcal{S}$. The procedure of finding the best neuron description can be formalized as:

$$\mathcal{E}_{\text{CompExp}}(f_i) = \arg\max_{s \in \mathcal{L} \subset \mathcal{S}} \delta_{\text{IoU}}\left(f_i, s\right). \tag{7}$$

Similar to NetDissect, CompExp requires datasets containing segmentation masks and is primarily applicable to convolutional neurons.

### A.1.3 MILAN

MILAN [13] is a method that aims to describe neurons within a DNN through open-ended natural language descriptions. First, a dataset of fine-grained human descriptions of image regions (Milannotations) is collected. These descriptions can be defined as concepts that are open-ended natural language descriptions, where $s \in \mathcal{N} \subset \mathcal{S}$. Given a DNN and input images $\boldsymbol{x} \in \boldsymbol{X}$, neuron masks $M(\boldsymbol{x}) \in \mathbb{R}^{H \times W \times C}$ are collected of highly activated image regions for $f_i$.

Two distributions are then derived: the probability $p(s|M(\boldsymbol{x}))$ that a human would describe an image region with $s$, and the probability $p(s)$ that a human would use the description $s$ for any neuron. The probability $p(s|M(\boldsymbol{x}))$ is approximated with the Show-Attend-Tell [48] image-to-text model trained on the Milannotations dataset. Additionally, $p(s)$ is approximated with a two-layer LSTM language model [50] trained on the Milannotations dataset.

These distributions are then utilized to find a description that has high pointwise mutual information with $M(\boldsymbol{x})$. A hyperparameter $\lambda \in \mathbb{R}$ adjusts the significance of $p(s)$ during the computation of pointwise mutual information (PMI) between descriptions $s$ and $M(\boldsymbol{x})$ sets, where the similarity $\delta_{\text{WPMI}}$ is weighted PMI (WPMI). The objective for WPMI is given by:

$$\delta_{\text{WPMI}}(s) = \log p\left(s|M(\boldsymbol{x})\right) - \lambda \log p(s). \tag{8}$$

MILAN aims to optimize high pointwise mutual information between $s$ and $M(\boldsymbol{x})$ to find the best description for $f_i$:

$$\mathcal{E}_{\text{MILAN}}(f_i) = \underset{s \in \mathcal{N} \subset \mathcal{S}}{\arg\max} \, \delta_{\text{WPMI}}\left(f_i, s\right). \tag{9}$$

The requirement of collecting the curated labeled dataset, Milannotations, limits MILAN's capabilities when applied to tasks beyond this specific dataset. Additionally, another drawback is the requirement for model training.

### A.1.4   FALCON

The FALCON [14] explainability method has a similar approach to MILAN. Initially, it gathers the most highly activating images corresponding to a neuron. GradCam [5] is subsequently applied to identify highlighted features in these images, which are then cropped to focus on these regions. These cropped images, along with large captioning dataset LAION-400m [51] with concepts $s \in \mathcal{N} \subset \mathcal{S}$, are input to CLIP (Contrastive Language-Image Pre-training) [27], which computes the image-text similarity between the text embeddings of captions and the input cropped images. The top 5 captions are then extracted. Conversely, the least activating images are collected, and concepts are extracted and removed from the top-scoring concepts, ultimately yielding the explanation of the neuron.

The similarity $\delta_{\text{CLIPScore}}$ is obtained by calculating the CLIP confidence matrix, which is essentially a Cosine Similarity matrix. The aim is to find the maximum image-text similarity score between image embeddings and their closest text embeddings from a large captioning dataset:

$$\mathcal{E}_{\text{FALCON}}(f_i) = \underset{s \in \mathcal{N} \subset \mathcal{S}}{\arg\max} \, \delta_{\text{CLIPScore}}\left(f_i, s\right). \tag{10}$$

This restriction significantly narrows down the range of models suitable for analysis, setting it apart considerably from other explanation methods.

**FALCON Implementation**   In its original implementation, FALCON restricts the set of "explainable neurons" based on specific parameters. These include the parameter $\alpha \in \mathbb{N}$, which determines the set of highly activating images for a given feature by requiring $\alpha > 10$. Additionally, it employs a threshold $\gamma \in \mathbb{R}$ for CLIP Cosine Similarity, with a set value of $\gamma > 0.8$.

These parameter settings significantly restrict the number of explainable neurons, resulting to fewer than 50 explainable neurons. This constraint prevents the necessary randomization for comparison with other methods. To address this, we set $\alpha = 0$ and $\gamma = 0$. However, for the original FALCON implementation, we retain the original settings of $\alpha$ and $\gamma$ and calculate $\Psi$ across all "explainable neurons". In our experiments on `ResNet18`, FALCON can only be applied to layers 2 to 4.

### A.1.5   CLIP-Dissect

CLIP-Dissect [52] is an explanation method that describes neurons in vision DNNs with open-ended concepts, eliminating the need for labeled data or human examples. This method integrates CLIP [27], which efficiently learns deep visual representations from natural language supervision. It utilizes both the image encoder and text encoder components of a CLIP model to compute the text embedding for

each concept $s \in \mathcal{C} \subset \mathcal{S}$ from a concept dataset and the image embeddings for the probing images in the dataset, subsequently calculating a concept-activation matrix.

The activations of a target neuron $f_i$ are then computed across all images in the probing dataset $\boldsymbol{X}$. However, as this process is designed for scalar neurons, these activations are summarized by a function that calculates the mean of the activation map over spatial dimensions. The concept corresponding to the target neuron is determined by identifying the most similar concept $s$ based on its activation vector. The most highly activated images are denoted as $\boldsymbol{X}_s \subset \boldsymbol{X}$.

SoftWPMI is a generalization of WPMI where the probability $p\left(\boldsymbol{x} \in \boldsymbol{X}_s\right)$ denotes the chance an image $\boldsymbol{x}$ belongs to the example set $\boldsymbol{X}_s$. Standard WPMI corresponds to cases where $p(\boldsymbol{x} \in \boldsymbol{X}_s)$ is either 0 or 1 for all $\boldsymbol{x} \in \boldsymbol{X}$, while SoftWPMI relaxes this binary setting to real values between 0 and 1. The function can be formalized as:

$$\delta_{\text{SoftPMI}}(s) = \log \mathbb{E}\left[p\left(s|\boldsymbol{X}_s\right)\right] - \lambda \log p(s). \tag{11}$$

The similarity function $\delta_{\text{SoftWPMI}}$ aims to identify the highest pointwise mutual information between the most highly activated images $\boldsymbol{X}_s$ and a concept $s$. This optimization search is expressed as:

$$\mathcal{E}_{\text{CLIP-Dissect}}(f_i) = \underset{s \in \mathcal{C} \subset \mathcal{S}}{\arg\max}\, \delta_{\text{SoftWPMI}}\left(f_i, s\right). \tag{12}$$

A drawback of CLIP-Dissect lies in its interpretability; descriptions are generated by the CLIP model, which itself is challenging to interpret.

### A.1.6  INVERT

Labeling Neural Representations with Inverse Recognition (INVERT) [16] shares the capability of constructing complex explanations like CompExp [12] but with the added advantage of not relying on segmentation masks and only needing labeled data. The method obtains its explanations by merging individual concepts into logical formulas using composition operators AND, OR, and NOT. It also exhibits greater versatility in handling various neuron types and is computationally less demanding compared to previous methods such as NetDissect [11] and CompExp [12]. Additionally, INVERT introduces a transparent metric for assessing the alignment between representations and their associated explanations. The non-parametric Area Under the Receiver Operating Characteristic (AUC) measure evaluates the relationship between representations and concepts based on the representation's ability to distinguish the presence from the absence of a concept, with statistical significance. The probing dataset with the concept present is labeled as $\boldsymbol{X}_1$, while the dataset without the concept is labeled as $\boldsymbol{X}_0$.

The goal of INVERT is to identify the concept $s \in \mathcal{L} \subset S$ that maximizes $\delta_{\text{AUC}}$ with the neuron $f_i$. Here, $s$ can be a combination of concepts. The optimization process resembles that of CompExp, employing beam search [49] to find the optimal compositional concept. The top-performing concepts are iteratively selected until the predefined compositional length $L \in \mathbb{N}$ is reached.

The similarity measure $\delta_{\text{AUC}}$ is defined as:

$$\delta_{\text{AUC}}(f_i) = \frac{\sum_{\boldsymbol{x}_0 \in \boldsymbol{X}_0} \sum_{\boldsymbol{x}_1 \in \boldsymbol{X}_1} \mathbf{1}[f_i(\boldsymbol{x}_0) < f_i(\boldsymbol{x}_1)]}{|\boldsymbol{X}_0| \cdot |\boldsymbol{X}_1|}. \tag{13}$$

The objective of INVERT is to maximize the similarity $\delta_{\text{AUC}}$ between a concept $s$ and the neuron $f_i$, which can be described as:

$$\mathcal{E}_{\text{INVERT}}(f_i) = \underset{s \in \mathcal{L} \subset \mathcal{S}}{\arg\max}\, \delta_{\text{AUC}}\left(f_i, s\right). \tag{14}$$

INVERT is constrained by the requirement of a labeled dataset and is computationally more expensive compared to CLIP-Dissect.

### A.2  Schematic Illustration of CoSy Implementation Details

In the example shown in Figure 1, we used the default settings of the explanation methods to generate explanations for neuron 80 in the avgpool layer of `ResNet18`. For CLIP-Dissect, we used the 20,000

most common English words as the concept dataset and the ImageNet validation dataset [36] as the probing dataset. We employed Stable Diffusion XL 1.0-base (SDXL) [34] as the text-to-image model, using the prompt "realistic photo of a close up of [concept]" to generate concept images, with [concept] being replaced by the textual explanation from the methods. We generated 50 images per concept for 50 randomly chosen neurons from the avgpool layer of ResNet18. For evaluation, we also used the ImageNet validation dataset as the control dataset.

## A.3    Prompt Bias Analysis

To address prompt bias and dataset dependency, we extended our analysis by comparing results on the object-focused ImageNet with the scene-focused Places365 dataset. Our results show a significant difference based on prompt selection and dataset: close-ups work well for object-centric datasets like ImageNet, while more general prompts, such as "photo of [concept]", are more suitable for scene-based datasets like Places356 (see Figure 7).

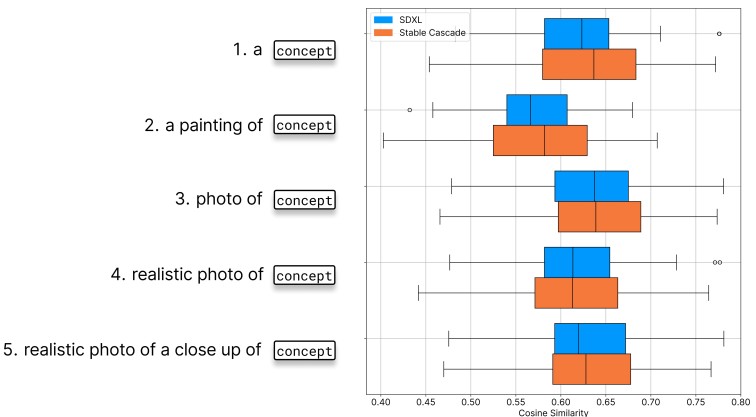

Figure 7: Cosine Similarity between synthetic and natural concept images for 50 concepts in the scene-based dataset Places365 [42], using different input prompts for the text-to-image models Stable Diffusion XL 1.0-base (SDXL) and Stable Cascade. The third prompt, "photo of [concept]", performs best for generating scene images.

Additionally, we evaluated model performance using different similarity metrics across both datasets. Alongside Cosine Similarity (*CS*), we introduced two more distance measures: Learned Perceptual Image Patch Similarity (LPIPS) [53], which calculates perceptual similarity between two images and aligns well with human perception, using deep embeddings from a VGG model. Additionally, we included "Euclidean Distance" (*ED*) to capture the absolute differences in pixel values. The results of these evaluations are presented in Table 5.

## A.4    Sanity Check Class Exclusion

In addition to the results presented in Table 2, we conducted the same experiment with the ground truth images excluded from the control dataset. The corresponding results, shown in Table 6, closely align with those obtained when the ground truth class was included. The findings are largely consistent with those obtained when the ground truth class was included.

## A.5    Intraclass Image Similarity

In addition to comparing natural and synthetic images as in Section 4.1, we also analyze the intraclass distance to compare the similarity among synthetic images. Intraclass distance refers to the degree of diversity or dissimilarity observed within a set of images of the same class. It quantifies how much the individual images deviate from the average or central tendency of the image set. In this context, intraclass distance is desirable, reflecting how visual concepts can appear in natural images. Higher similarity scores indicate greater divergence from natural occurrences of concepts.

Cosine Similarity (*CS*), "Euclidean distance" (*ED*), and Learned Perceptual Image Patch Similarity (LPIPS) are commonly used metrics for measuring image similarity because they capture different

Table 5: Synthetic-to-Natural Image Similarity. This table illustrates the impact of varying parameters within our method pipeline. We evaluate five different prompts as input to two different text-to-image models, Stable Diffusion XL 1.0-base (SDXL) and Stable Cascade (SC). We randomly selected 50 from both ImageNet and Places365, using each class name as input to the prompt, denoted as [concept], generating 50 images per prompt for each model. We compute the average similarity metrics across all classes and report the standard deviation. Higher values of *CS* (Cosine Similarity) and lower values of *ED* ("Euclidean Distance") and LPIPS (Learned Perceptual Image Patch Similarity) indicate greater similarity between synthetic and natural images. High similarity is desirable; **bold** values represent the highest scores for each prompt.

| Dataset | Prompt | Text-to-image | Similarity Measure | | |
| --- | --- | --- | --- | --- | --- |
| | | | *CS* ($\uparrow$) | *ED* ($\downarrow$) | LPIPS ($\downarrow$) |
| ImageNet | 1. "a [concept]" | SDXL | 0.70±0.08 | 8.0±1.09 | **0.72±0.03** |
| | | SC | 0.69±0.08 | 8.19±1.02 | 0.74±0.04 |
| | 2. "a painting of [concept]" | SDXL | 0.64±0.07 | 8.74±0.81 | 0.73±0.03 |
| | | SC | 0.64±0.07 | 8.81±0.76 | 0.76±0.04 |
| | 3. "photo of [concept]" | SDXL | 0.71±0.08 | **7.94±1.11** | **0.72±0.03** |
| | | SC | 0.69±0.08 | 8.25±1.01 | 0.75±0.04 |
| | 4. "realistic photo of [concept]" | SDXL | 0.70±0.08 | 8.07±1.09 | **0.72±0.03** |
| | | SC | 0.69±0.08 | 8.34±0.99 | 0.73±0.03 |
| | 5. "realistic photo of a close up of [concept]" | SDXL | **0.72±0.08** | 7.97±1.11 | **0.72±0.03** |
| | | SC | 0.69±0.08 | 8.31±1.01 | 0.73±0.04 |
| Places365 | 1. "a [concept]" | SDXL | 0.62±0.09 | 9.02±1.04 | **0.71±0.02** |
| | | SC | 0.63±0.09 | 8.96±1.04 | 0.72±0.03 |
| | 2. "a painting of [concept]" | SDXL | 0.57±0.08 | 9.51±0.83 | 0.72±0.02 |
| | | SC | 0.58±0.08 | 9.43±0.83 | 0.72±0.03 |
| | 3. "photo of [concept]" | SDXL | **0.64±0.09** | **8.85±1.09** | **0.71±0.02** |
| | | SC | **0.64±0.09** | 8.89±1.08 | 0.72±0.03 |
| | 4. "realistic photo of [concept]" | SDXL | 0.62±0.09 | 9.08±1.02 | **0.71±0.02** |
| | | SC | 0.62±0.08 | 9.21±0.98 | 0.72±0.03 |
| | 5. "realistic photo of a close up of [concept]" | SDXL | 0.63±0.09 | 9.04±1.05 | 0.71±0.03 |
| | | SC | 0.63±0.09 | 9.14±1.01 | 0.72±0.03 |

aspects of similarity and complement each other. We compute the average *CS*, *ED*, and LPIPS for each class and determine the overall class average. Table 7 provides a detailed overview of the results quantifying the similarity within synthetic images using *CS* and *ED*. When evaluating these results, it is important to note that high scores do not necessarily indicate optimal outcomes, as they suggest nearly identical images, which may lack intraclass distance. Conversely, very low scores imply significant differences among images, which might not capture the essence of the concept adequately. Ideally, we aim for somewhat similar yet slightly varied images representing the same class. The results show that the Stable Cascade (SC) model consistently achieves higher scores across all prompts compared to the Stable Diffusion XL 1.0-base (SDXL) model. Notably, it obtains the highest score for the two most elaborate prompts (4, 5). This indicates that the SC model tends to offer less intraclass distance in visually representing concepts.

Table 6: Comparison of true and random explanations on output neurons with known ground truth labels. This table presents the average quality scores (with standard deviation) for true explanations, derived from target class labels, and random explanations, derived from randomly selected synthetic image classes (excluding the target class), across four models pre-trained on ImageNet. Higher values are better. Our results consistently show high scores for true explanations and low scores for random ones.

| Model | AUC ($\uparrow$) | | MAD ($\uparrow$) | |
| --- | --- | --- | --- | --- |
| | True | Random | True | Random |
| ResNet18 | 0.98±0.09 | 0.52±0.24 | 6.59±2.14 | 0.08±0.91 |
| DenseNet161 | 0.99±0.08 | 0.52±0.24 | 7.33±1.91 | 0.04±0.84 |
| GoogLeNet | 0.99±0.07 | 0.49±0.24 | 8.01±2.28 | -0.01±0.81 |
| ViT-B/16 | 0.99±0.05 | 0.52±0.22 | 14.7±3.88 | 0.13±1.11 |

Table 7: Intraclass Image Similarity. This table illustrates the impact of varying parameters within our COSY framework. We evaluate five different prompts as input to two different text-to-image models. A random selection of 50 ImageNet classes is made, with each class name used as input to the prompt, denoted as [concept], resulting in 50 images generated per prompt using a text-to-image model. We compute the average intraclass similarity across all classes and report the standard deviation. Higher *CS* and lower *ED* values indicate greater similarity between the images. In intraclass image similarity, neither excessively high nor excessively low scores are desirable.

| Dataset | Prompt | Text-to-image | Similarity Measure | | |
| --- | --- | --- | --- | --- | --- |
| | | | *CS* ($\uparrow$) | *ED* ($\downarrow$) | LPIPS ($\downarrow$) |
| ImageNet | 1. "a [concept]" | SDXL | 0.85±0.07 | 5.61±1.40 | 0.61±0.11 |
| | | SC | 0.93±0.03 | 3.76±0.92 | 0.49±0.10 |
| | 2. "a painting of [concept]" | SDXL | 0.87±0.05 | 4.94±1.13 | 0.62±0.10 |
| | | SC | 0.93±0.03 | 3.67±0.87 | 0.54±0.10 |
| | 3. "photo of [concept]" | SDXL | 0.84±0.06 | 5.65±1.34 | 0.62±0.10 |
| | | SC | 0.92±0.04 | 4.03±0.99 | 0.52±0.10 |
| | 4. "realistic photo of [concept]" | SDXL | 0.87±0.06 | 5.15±1.31 | 0.60±0.10 |
| | | SC | 0.94±0.03 | 3.53±0.86 | 0.45±0.09 |
| | 5. "realistic photo of a close up of [concept]" | SDXL | 0.89±0.04 | 4.82±1.16 | 0.60±0.10 |
| | | SC | 0.94±0.03 | 3.58±0.86 | 0.50±0.09 |
| Places365 | 1. "a [concept]" | SDXL | 0.84±0.06 | 5.75±1.35 | 0.61±0.09 |
| | | SC | 0.92±0.03 | 4.02±0.95 | 0.54±0.09 |
| | 2. "a painting of [concept]" | SDXL | 0.88±0.04 | 4.89±1.05 | 0.61±0.09 |
| | | SC | 0.93±0.03 | 3.79±0.84 | 0.56±0.09 |
| | 3. "photo of [concept]" | SDXL | 0.85±0.06 | 5.61±1.27 | 0.61±0.10 |
| | | SC | 0.93±0.03 | 3.94±0.92 | 0.54±0.09 |
| | 4. "realistic photo of [concept]" | SDXL | 0.88±0.05 | 5.05±1.15 | 0.59±0.09 |
| | | SC | 0.94±0.03 | 3.70±0.88 | 0.52±0.08 |
| | 5. "realistic photo of a close up of [concept]" | SDXL | 0.87±0.05 | 5.25±1.24 | 0.60±0.09 |
| | | SC | 0.94±0.03 | 3.68±0.91 | 0.54±0.09 |

## A.6 Model Stability

In this experiment, our goal is to evaluate the stability of the image generation method employed, aiming to ensure consistent results within our COSY framework. We achieve this by varying the seed of the image generator and observing the impact on image generation. We anticipate consistent image representations across different model initializations, thus ensuring the stability of our framework.

For our analysis, we utilize ResNet18 and focus on its output neurons, as the ground-truth labels associated with these neurons are known. We randomly select six classes $s$ from the ImageNet validation dataset [36] and examine the corresponding $f_i$ class output neurons using COSY. Here, we exclude the $s$ class from $\mathbb{A}_0$ and let $\mathbb{A}_1$ represent the $s$ class. To ensure robustness, we initialize the text-to-image model across a random set of 10 seeds. Our analysis involves calculating the first (mean) and second moment (STD) using $\Psi_{\text{AUC}}$, as well as evaluating the intraclass image similarity (refer to Section A.5) within each synthetic ground truth class.

The results for our experiment, as shown in Table 8, demonstrate remarkably high AUC scores, indicating near-perfect detection of synthetic ground truth classes across all image model initializations. Furthermore, the standard deviation is exceptionally low, suggesting consistent image generation regardless of the chosen seed. The intraclass similarity values indicate a certain degree of distance in the generated images, indicating high similarity yet distinctiveness. This intraclass distance is desirable, ensuring that the images are not identical but share common characteristics.

These findings underscore the reliability and consistency of our image generation pipeline within our COSY framework. The high stability of text-to-image generation across different seeds and the diversity of image similarity contribute to the robustness of our approach.

## A.7 Compute Resources

For running the task of image generation for COSY we use distributed inference across multiple GPUs with PyTorch Distributed, enabling image generation with multiple prompts in parallel. We run our script on three Tesla V100S-PCIE-32GB GPUs in an internal cluster. Generating 50 images for 3 prompts in parallel takes approximately 12 minutes.

Table 8: Model Stability. A comparison of various model initializations across 10 random seeds using SDXL. The results represent the average scores with standard deviations for each class, calculated across 10 seeds.

| Concept | AUC (↑) | Similarity Measure | | |
| --- | --- | --- | --- | --- |
| | | *CS* (↑) | *ED* (↓) | LPIPS (↓) |
| bulbul | 0.9996±0.0002 | 0.91±0.03 | 3.99±0.66 | 0.65±0.06 |
| china cabinet | 0.9999±0.0001 | 0.89±0.04 | 5.00±0.90 | 0.58±0.04 |
| leatherback turtle | 0.9994±0.0001 | 0.91±0.04 | 4.65±0.87 | 0.63±0.04 |
| beer bottle | 0.9919±0.0038 | 0.80±0.08 | 6.79±1.41 | 0.69±0.08 |
| half track | 0.9998±0.0000 | 0.88±0.04 | 5.12±0.91 | 0.61±0.04 |
| hard disc | 1.0000±0.0001 | 0.90±0.05 | 4.64±1.17 | 0.60±0.05 |
| **Overall Mean** | **0.9984±0.0007** | **0.88±0.02** | **5.03±0.26** | **0.63±0.05** |

## A.8 Additional Results for Method Comparison across ResNet18 Layers

Given that the original implementation of FALCON only provides results for their defined "explainable neurons" (see Appendix A.1.4), we included additional results comparing all methods based on this subset of neurons. Specifically, there are 7 explainable neurons in layer 2, 5 in layer 3, and 15 in layer 4. Figure 8 presents these results.

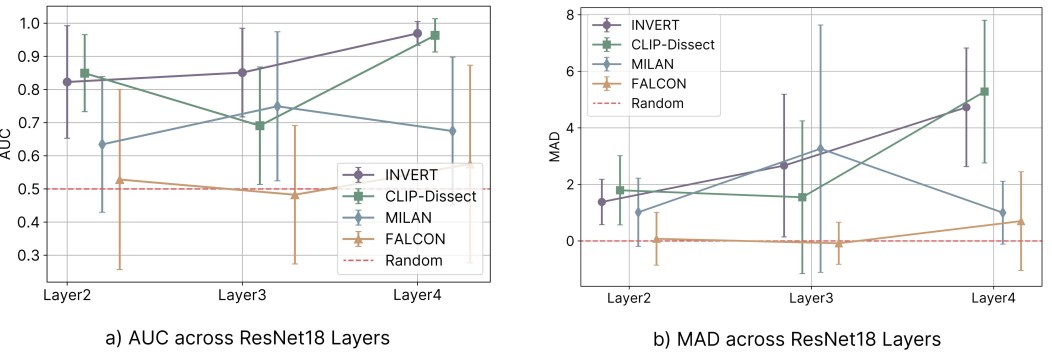

a) AUC across ResNet18 Layers

b) MAD across ResNet18 Layers

Figure 8: A comparison of explanation methods in ResNet18 shows that INVERT and CLIP-Dissect maintain high MAD scores across all layers, while MILAN and FALCON have lower scores. Overall, performance declines in the lower layers for all methods.

## A.9 Qualitative Examples of Lower-Level Concepts

The performance of generative models may present challenges when dealing with abstract concepts. To assess this concern and provide a basis for further discussion, we have included an additional example of lower-layer concepts in Figure 9.

## A.10 Concept Broadness

While COSY focuses on measuring the explanation quality, another open question is how broad or abstract are the concepts provided as textual explanations. This question of how specific or general an individual neuron is described by the explanation, might be relevant to different interpretability applications. For example, research fields where the user aims to deploy the same network for multiple tasks with varying image domains. In this case, describing a neuron's more general concept such as "a round object" might be more informative than a more (domain-)specific concept such as "a tennis ball" for the network assessment. To provide insight into the broadness of concepts, we assessed whether the similarity between images generated based on the same concept changes from more general to more specific concepts.

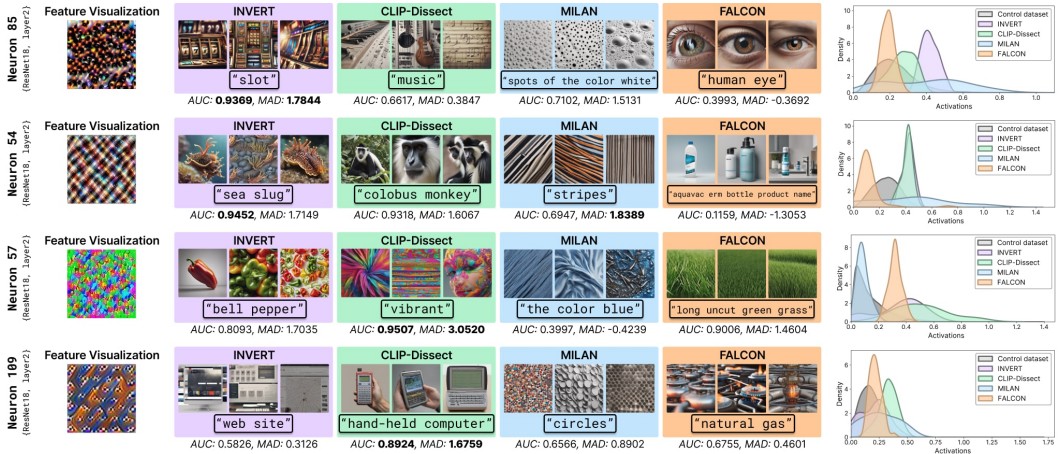

Figure 9: Examples of various explanations from four different methods applied to the low-layer neurons in layer 2 of the `ResNet18` model are provided. From the illustrated Feature Visualizations, we can observe that these neurons detect low-level abstractions. However, the methods studied generally fail to provide low-level explanations, instead attributing more complex explanations.

In our experiment, we define the broadness of a concept based on the number of hypernyms in the WordNet hierarchy [54]. The more specific a concept the larger the number of hypernyms. We choose two ImageNet classes ("ladybug", "pug") and generate 50 images for each concept as well as each hypernym of both concepts (with the most general concept being "entity"). Then, we measure the Cosine Similarity (*CS*) of all images generated based on the same concept. The box plot of the *CS* across both concepts and all hypernyms, in Figure 10 indicates that we do not find a correlation. Thus, we hypothesize that the chosen temperature of the diffusion model has a stronger effect on image similarity than the broadness of the prompt used for image generation.

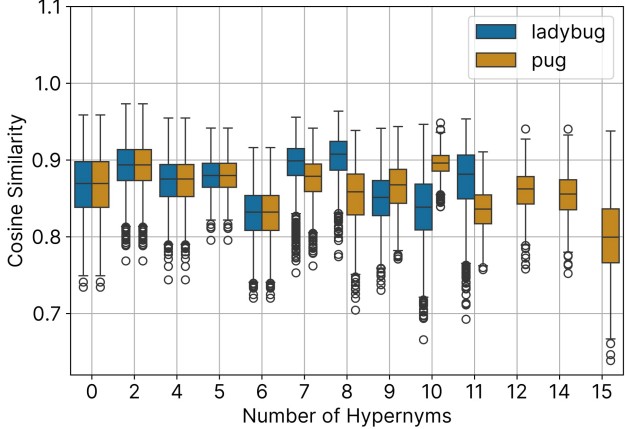

Figure 10: The figure demonstrates the independence of the concept broadness measured by the number of hypernyms as defined in WordNet [54] to the inter-image similarity of corresponding generated images.

## A.11 Prompt and Text-to-Image Model Comparison

Figure 11 showcases additional examples of synthetically generated images using both `SDXL` and `SC` across various prompts, highlighting the diversity and accuracy of concept representation.

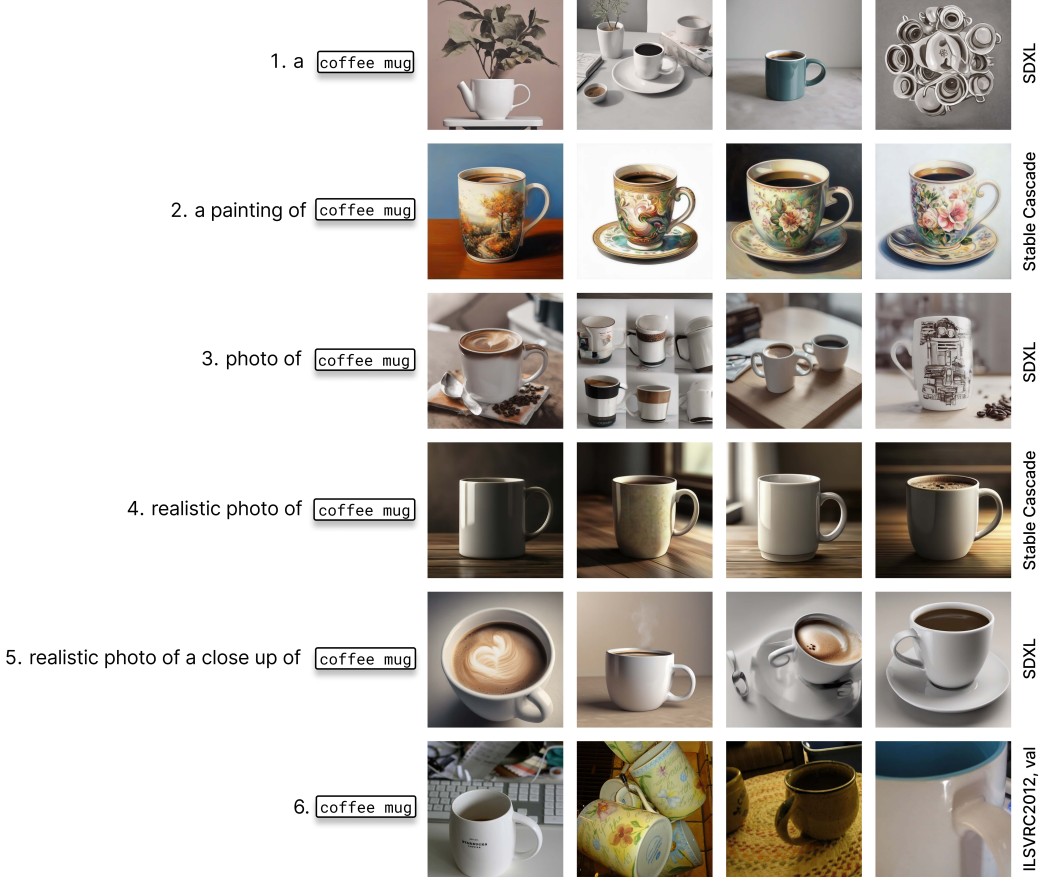

Figure 11: Example images for "coffee mug" generated by the text-to-image models SDXL and SC across various prompts. (1) and (3) present examples of synthetic images with relatively low intraclass similarity and relatively high natural-to-synthetic similarity scores. (2) shows examples of synthetic images with the lowest similarity to natural images. (4) illustrates examples of synthetic images with the highest similarity to other synthetic images within the same class. (5) showcases examples of synthetic images with the highest similarity to natural images. (6) displays examples of natural images from the ImageNet validation dataset [36] belonging to the class "coffee mug".

