# OpenReview forum: "CoSy: Evaluating Textual Explanations of Neurons"
_NeurIPS.cc/2024/Conference — NeurIPS 2024 poster_

### Official Review · Reviewer_9Kbo · 2024-06-25

**Soundness:** 4
**Presentation:** 4
**Contribution:** 4
**Rating:** 7
**Confidence:** 4

**Summary:**

The authors propose a framework for evaluating Neuron Annotation methods which label a neuron in a given vision model, via a textual description. This framework is based on generating a set of images using a text2image model, given the predicted textual description of the neuron which acts as the textual input prompt to the generative model. The idea is that if the textual description truly describes the neuron, then the generated image should activate that neuron more than that of a random baseline image from the training dataset the model was trained on (denoted as the control image). The authors use two scoring functions: the AUC and MAD for quantitative evaluation. The authors further evaluate the effectiveness of the evaluation framework, showing that it is a valid way. They then proceed to evaluating existing Neuron Annotation works, and drawing findings and conclusions, such as the fact that some methods are generating random textual descriptions of neurons.

**Strengths:**

I would like to congratulate the authors for their work. I really enjoyed reading the paper.
- This work is the first to propose an unified, intuitive and valid way of evaluating neuron annotation works. Indeed, most of these works provide their own evaluation metrics which makes it hard to compare with previous works, and verify if one method is better than the other
- The paper is very well written, understandable and straight-forward. The evaluation framework is also simple, something which all people and researchers will appreciate.
- The papers provides a meta-evaluation section to show the validity of the proposed framework, which I find interesting and important
- The finding that some Neuron Annotation works provide wrong predictions of textual descriptions (or let's say, statistical-based descriptions) is very interesting and important, which raises a concern in the interpretability field, especially in safety-critical applications.

**Weaknesses:**

I could not really find major weaknesses that are grounds for rejection. There are some moderate weaknesses:

- [W1] How does the proposed evaluation framework relate to the other evaluation methods used in existing Neuron Annotation works? Do they align well with the new proposed measure? For example, does the proposed evaluation framework align well with the BERTScores for the human annotations of MILAN? In CLIP-Dissect, they labeled the neurons of the final classification layer and measured accuracy with the ground-truth class names. Does the proposed evaluation framework align well here also?
- [W2] The pool of models is a bit small. More models (especially for for ImageNet) should be analysed. ImageNet ResNet50 is a common model in most Neuron Annotation works and should be reported. What about self-supervised models, such as DINO? How do they compare to classifiers? MILAN also performs analysis on these models.
- [W3] There is another related Neuron Annotation work [R1], which is only applicable to ViTs. Here, attention heads and tokens are considered as Neurons and labeled. The authors may extend the analysis to this work to strengthen the paper, given that [R1] is a mathematically-valid decomposition of the ViT, and therefore should align best with the proposed evaluation framework compared to other Neuron Annotation works using the ViT.
- [W4] The Area Under the Receiver Operating Characteristic (AUC) in section 3.2 seems to be a measure which does not involve generating a curve. But usually AUC means that we have a curve in the first place before taking the area under it. What is the curve in the author's case?

Other minor weaknesses:
- The text2image models are very time-consuming (the authors also mention that it takes 12 min using a parallel GPU setup, for generating 50 images). This limits the applicability to new users that want to evaluate their works. It would be nice to report the speed of different Stable Diffusion models along in Section 4.1, and include other fast-optimized and less-computationally expensive models -- i am not an expert in the text2image field so i do not know which models in the current literature are faster than the ones the authors use. Fortunately, given how fast the generative text2image field is moving, this problem will not be a problem in the very near future.
- The fact that CLIP-dissect achieves good results in not surprising - because CLIP-Dissect interprets CLIP Neurons rather than CNN/ViT neurons. Their method is based on scaling CNN/ViT neuron activations using CLIP. Therefore, any image-text pair judged low by CLIP will also reduce the CNN/ViT neuron activation value for the corresponding image. In essence, one could think of CLIP-dissect as an amplified value of the CLIP image-text score (amplified by the CNN/ViT neuron activation value for the same image). And since CLIP itself is a very strong model and especially in scoring image-text inputs, it is not surprising that is performs best.
- Line 156, "synthetic" should be replaced by "natural images" (as you already used "generated images" in line 155). Also, please be consistent with terminology.

[R1] INTERPRETING CLIP’S IMAGE REPRESENTATION VIA TEXT-BASED DECOMPOSITION, ICLR 2024

**Questions:**

In general, i feel this paper is a clear accept to me, and it will have a good impact in the field. I would like that the moderate weaknesses are addressed. While I understand that Rebutal time is limited, and computation time of the proposed framework is also time-consuming, I encourage the authors to address as much as they can from the moderate weaknesses (W1-W2-W3-W4).

The minor weaknesses are just comments that can be included in the final manuscript.

**Limitations:**

Limitations are discussed. No special issues with negative societal impact.

---

> ### Author Rebuttal · Authors · 2024-08-07
>
> We appreciate Reviewer 4 (R4) for their time taken and their great attention to detail shown in the review. We are honored by your positive feedback, and thankful that you consider our work as important.
>
> **[A1] Comparison to other evaluation scores:**
> We thank the reviewer for asking these important questions. We extend the discussion of comparable methods in the final manuscript. While the results presented in the primary publication are comparable, with BERTScores around 0.5 for MILAN across architectures and good performance of CLIP-Dissect, we argue that due to the lack of standardized procedures across publications, we cannot rule out biases toward specific methods. Moreover, human-based evaluation methods often also include further biases. For example, many evaluations supply highly activating images [1], to make the neuron annotation by humans possible. These images may not accurately represent the neuron's overall behavior by only referencing the maximum tail of the distribution.
>
> **[A2] Broader range of explained models:**
> We acknowledge the reviewer's suggestion to analyze a broader range of models. In response, we have now included ResNet50 pretrained on ImageNet in our analysis as suggested (see Table 2 in PDF). We can observe that INVERT achieves the highest AUC score, while CLIP-Dissect attains the highest MAD score, consistent with our previous benchmarking results on ResNet50 pretrained on Places365. Additionally, we plan to incorporate additional models, such as the suggested DINO, in our benchmarking table for a broader evaluation.
>
> **[A3] Additional textual explanation methods:**
> We highly value the reviewer's comment and suggestion to include more explanation methods. Due to the limited rebuttal period, we were not able to complete the additional computations in time. Nonetheless, we aim to address this request going forward.
>
> **[A4] Clarification on AUC:**
> Thank you for your remark on the AUC. We apologize for not describing it clearly enough in our initial submission. To clarify, we have included a plot in Figure 3 in the PDF that illustrates the AUC scores for a specific neuron (Neuron 358, ResNet18, Layer 4) across four different methods.
> The AUC score is the Area Under the Receiver Operating Characteristic (ROC) Curve. Figure 3 in PDF illustrates these ROC curves. In this context, the target is our binary label (0 for the control dataset and 1 for concept images), and the ''predictions'' are the neuron activations for each image. The AUC demonstrates how well the activations of the concept images discriminate against the control image activations.
> In our benchmarking table, we present the mean AUC scores across all selected neurons, providing a more comprehensive evaluation. We hope this explanation and the included visualizations help to clarify our use of AUC.
>
> **[A5] Computational Cost:**
> We strongly agree with R4 that the computational cost can be a limiting factor. Thus, we tried to incorporate other open-source text-to-image models, which claim to improve upon computation time but found no significant improvements. In case of limited resources, the only option therefore is the reduction of inference steps to improve run-time. We note, however, that a smaller number of inference steps reduces image quality, which might affect the algorithmic performance.
>
> **[A6] CLIP Bias:**
> We highly appreciate this question. The explanations from the CLIP-Dissect method are selected from a predefined list of concepts, therefore the potential bias towards CLIP-Dissect is minimal.
>
> **[A7] Consistent Terminology:**
> We thank the reviewer for pointing out the inconsistency and will ensure consistent terminology in the final manuscript.
>
> [1] Kalibhat, Neha, et al. ''Identifying interpretable subspaces in image representations.'' ICML, 2023.

---

> > ### Comment · Reviewer_9Kbo · 2024-08-08
> > **Response to rebuttal**
> >
> > I thank the authors for the rebuttal and the extra experiments they have conducted. I am generally happy and positive about this paper, and I will retain my score as an "Accept". The authors should, however, incorporate these extra experiments, as well as other reviewers comments, in the revised manuscript.
> >
> > Regarding [A6], i was referring to the bias that CLIP-Dissect method carries towards CLIP (it explains CLIP rather than other models such as ResNet-50 due to the reasons I mentioned). I was not referring to the authors's method bias. This was not really a weakness, rather a justification of a finding of why CLIP-Dissect performs better.
> >
> > I wish the authors the best of luck and congratulations!

---

> > > ### Author Response · Authors · 2024-08-14
> > >
> > > Thank you for your positive feedback and support. We agree that the additional experiments strengthen the paper, and we are already conducting more, which we will include in the revised manuscript along with the reviewers' comments.
> > >
> > > Regarding [A6], we understand now that your comment was about the inherent bias of the CLIP-Dissect method towards explaining CLIP models, rather than towards the predefined concepts. Thank you for the clarification.
> > >
> > > Thank you again for your insights and encouragement!

---

### Official Review · Reviewer_yev1 · 2024-06-26

**Soundness:** 3
**Presentation:** 3
**Contribution:** 1
**Rating:** 6
**Confidence:** 4

**Summary:**

The authors present a new framework designed to evaluate the quality of textual explanations for neurons in deep neural networks (DNN). To this end, the paper introduces CoSY, which aims to provide a quantitative, architecture-agnostic evaluation method for these explanations, addressing the challenge of the lack of unified, general-purpose evaluation metrics in the field. Given a neuron $f$ of a DNN and explanation $s \in S$, CoSY evaluates the alignment between the explanation and a neuron in three simple steps. It utilizes a generative model to create synthetic data points from textual explanations and compares the neuron's response to these data points with its response to control data points. This comparison provides a quality estimate of the textual explanations. Further, they perform a large set of empirical analyses to quantify the effectiveness of their proposed AUC and MAD metrics.

**Strengths:**

1. The paper introduces a new quantitative framework that allows comparisons of different explanation methods, facilitating a more standardized approach to evaluating the quality of neuron explanations​​.

2. The proposed framework is independent of specific neural network architectures, making it broadly applicable to various models in computer vision and potentially other domains.

3. The paper includes extensive meta-evaluation experiments to validate the reliability of CoSY.

**Weaknesses:**

1. The authors indicate that "the results using prompt 5 as input to SDXL leads to the highest similarity to natural images" --- isn't there an implicit bias towards images with a single object in these results? Given that, most of the images in datasets like ImageNet consist of single objects, generating synthetic images using a concept and comparing them with natural images should only work for clean natural images, where the concept is explicitly observable.

2. The authors argue that "different methods devised their evaluation criteria, making it difficult to perform general-purpose, comprehensive cross-comparisons." While it is true that the community should work towards common benchmark metrics, it would be great if the authors could clarify what is the problem with the existing quantitative metrics. Further, why can't someone benchmark existing metrics?

3. In Lines 159-161, the authors mention that they employ cosine similarity (CS) to measure the similarity between synthetic images and natural images corresponding to the same concept, but they never discuss the problems or motivations of using CS. For instance, cosine similarity is not a well-calibrated metric (also observed in Fig 2 where we observe very little variance in cosine values across different prompts) for models like CLIP, where the representations are densely packed using a large number of concepts.

4. Can the authors comment how possible techniques to alleviate the dependency of a text-to-image generative model in their evaluation pipeline?

**Questions:**

1. Should we expect the AUC and MAD evaluation metrics to correlate? i.e., will an explanation method achieving high AUC also obtain a high MAD score? If yes, why don't we observe this consistently in the empirical analysis?

2. Is quantifying the synthetic image reliability using only 10 random concepts from the 1,000 classes of ImageNet in Figure 2 justified? Do these results hold for a larger number of concepts?

3. Is the aggregation of the activation values done using absolute sum or signed sum of the activations? This is important as negative activations may bias the aggregate value and impact the AUC and MAD metrics.

4. The authors present a very interesting analysis in Fig 4, where they aim to study the quality of explanations for neurons in different layers of a model. However, the error in the values in Fig. 4 is very large, questioning the conclusions from the figure. Moreover, given that the first few layers of a model detect generic low-level features like corners, edges, angles, and colors, why don't we observe near-perfect cosine similarity?

**Limitations:**

Please refer to the weakness section for more details.

---

> ### Author Rebuttal · Authors · 2024-08-07
>
> We thank Reviewer 3 for the constructive and insightful remarks and appreciate the positive feedback regarding the presentation quality. Below, we address each of their points in detail.
>
> **[A1] Prompt Bias:** We appreciate and strongly agree with the reviewer's insight regarding the dataset dependency of the prompt. We have addressed this in our manuscript by comparing the ImageNet dataset (object-focused) with the Places365 (scene-focused), leading us to use a more general prompt for Places365 models.
> For further investigations, we expanded our image similarity analysis from 10 to 50 classes, as suggested in [Q2]. We also compared results across both datasets on different similarity measures, as suggested in [W3]. For a detailed analysis, please refer to Figure 2 and Table 1 in our PDF. Our results show a significant difference based on prompt selection and dataset: close-ups work well for object-focused datasets, while more general prompts like ''photo of'' are better for scene datasets.
>
> **[A2] Prior Evaluation Methods:**
>
> Benchmarking existing metrics is challenging due to several reasons:
>
> - **Challenges in Human Evaluations:** Human evaluations mostly fail to give a holistic description of a neuron, since they rely on describing highly activating images only [1]. Furthermore, human-based evaluations suffer from a lack of standardized experimental setups, e.g. varying protocols, tasks, and participant groups, introducing biases, inconsistencies, and are overall not scalable.
>
> - **Limited Scope of Ground-Truth Labels:** Label-based evaluations are restricted to output neurons with predefined classes [2, 3]. Intermediate layers and open-vocabulary explanations are not covered, which limits the applicability of these metrics for a comprehensive evaluation across all neurons in a model. Furthermore, although output neurons are trained to detect specific concepts, their performance may not exactly match their intended function.
>
> We will add these points to the paper. Thus, clarifying that our approach unifies the evaluation procedure for open-vocabulary explanations and future textual explanation methods for any neuron.
>
> **[A3] Limitations of Cosine Similarity as image similarity measure:**
> We acknowledge the reviewer's concern regarding the limitations of using cosine similarity (CS). In response, we have incorporated two additional distance measures: Learned Perceptual Image Patch Similarity (LPIPS), which calculates perceptual similarity between two images and aligns well with human perception, using deep embeddings from a VGG model. Additionally, we included Euclidean Distance (ED) to capture the absolute differences in pixel values.
> These measures provide further context, allowing a broader assessment of visual similarity beyond CS. For detailed results and comparisons, please refer to our PDF, particularly Table 1, which analyzes image similarity with more concepts, as suggested in [Q2]. Importantly, our overall conclusions remain consistent even with these new metrics.
>
> **[A4] Dependency of a text-to-image generative model in evaluation framework:**
> For our proposed approach, txt2img models are crucial, because most explanation methods rely on open-vocabulary descriptions for which images are not always available.
>
> Theoretically, a way to mitigate reliance on txt2img models is to manually collect data points corresponding to the neuron label (i.e., images that, in the CoSy approach, are generated by the txt2img model), which however in practice would not be scalable. Furthermore, we ensure that the effects of the chosen txt2img model are limited by performing the sanity checks (see Section 4).
>
> **[Q1] Correlation of AUC and MAD:**
> CoSy measures the difference between activation distributions in the control dataset and the images corresponding to the given explanations. The AUC, a nonparametric test, assesses whether a neuron ranks data points corresponding to the explanation systematically higher than random images. In contrast, MAD is a parametric test similar to the Student's t-test, using exact activations and being more susceptible to outliers. While practically high MAD scores often accompany high AUC scores, these metrics can sometimes disagree, thus complementing each other and providing a broader evaluation of the explanation's quality.
>
> **[Q2] Small Concept Size:**
> We agree and expanded our experiment to include 50 ImageNet concepts, as shown in Figure 2 of the PDF, and applied the same methodology to 50 scene concepts from the Places365 dataset. The results for ImageNet remain consistent, with increased standard deviation but maintaining the general trend.
>
> **[Q3] Aggregation operation:**
> The aggregation operation is only performed when the output of a specific neuron is multidimensional (non-scalar) and follows Equation 2 in the paper, corresponding to the average pooling operation. Negative activations do not impact the AUC or MAD metrics because these metrics assess distribution differences.
>
> **[Q4] Large errorbars in Figure 4:**
> We acknowledge the concern about the large errors in Figure 4. These errors stem from the evaluation methods' performance. Contrary to the reviewer's claim, Figure 4 does not measure cosine similarity (CS); it measures AUC and MAD. The significant errors suggest that no single method is definitively superior, but the results still offer valuable insights for future research and improvements.
>
> [1] Kalibhat, Neha, et al. ''Identifying interpretable subspaces in image representations.'' ICML, 2023.
>
> [2] Bykov, Kirill, et al. ''Labeling Neural Representations with Inverse
> Recognition.'' NeurIPS, 2024.
>
> [3] Oikarinen, Tuomas, et al. ''CLIP-Dissect: Automatic Description of Neuron Representations in Deep Vision Networks.'' ICML, 2023.

---

> > ### Comment · Reviewer_yev1 · 2024-08-08
> >
> > Thank you for your detailed rebuttal response. Your responses have clarified most of my concerns. I increase my score to "Weak Accept".

---

> > > ### Author Response · Authors · 2024-08-08
> > >
> > > Thank you for taking the time to review our rebuttal and for your feedback. We're glad we could address your concerns and sincerely appreciate your revised assessment.

---

### Official Review · Reviewer_hkvS · 2024-07-08

**Soundness:** 3
**Presentation:** 3
**Contribution:** 3
**Rating:** 7
**Confidence:** 4

**Summary:**

The paper presents a novel, architecture-agnostic framework called COSY for quantitatively evaluating textual explanations of neurons in deep neural networks. The framework utilizes generative models to create synthetic images based on textual explanations, allowing for a standardized comparison of neuron responses to these synthetic images versus control images. Through a series of meta-evaluation experiments, the authors demonstrate the reliability and practical value of COSY by benchmarking various concept-based textual explanation methods for computer vision tasks, revealing significant variability in their quality.

**Strengths:**

1. The problem of evaluating textual explanations of neurons is worth studying, which is critical for the advancement of explainable AI and the wider adoption of machine learning models.

2. The COSY framework is designed to be architecture-agnostic, meaning it can be applied to any computer vision model regardless of its underlying architecture.

3. COSY introduces a novel, quantitative evaluation framework for textual explanations of neurons, which addresses the lack of unified, general-purpose evaluation methods.

4. Through the COSY framework, various existing concept-based textual explanation methods can be benchmarked and compared. This is demonstrated in the paper by benchmarking multiple explanation methods, revealing significant differences in their quality and providing insights into their performance.

**Weaknesses:**

1. The effectiveness of the COSY framework relies heavily on the quality of the generative models used to create synthetic images. If the generative models are not trained on a diverse set of concepts, they may fail to produce accurate synthetic images, thereby affecting the evaluation's reliability.

2. As mentioned by the authors, the INVERT method optimizes for the AUC metric during explanation generation; it may be biased towards achieving higher AUC scores in the COSY evaluation, potentially leading to an overestimation of its performance compared to other methods.

3. As noted in the paper, the COSY framework demonstrates a decline in explanation quality for neurons in the lower layers of the network, which typically encode lower-level features. Could the authors provide more insight into this phenomenon? Is it possible that this is due to limitations in the generative models' capabilities, given that lower layers generally encode more basic concepts which might not be well-represented by the generative model?

**Questions:**

see the Weaknesses part

**Limitations:**

Yes, the authors adequately addressed the limitations.

---

> ### Author Rebuttal · Authors · 2024-08-07
>
> We thank Reviewer 2 (R2) for the detailed comments and in-depth remarks. We are pleased that our work was found to be a valuable contribution. Below, we address all individual comments.
>
> **[A1] Text-to-image models:** We agree that the effectiveness of CoSy depends up on the performance of the generative model, which indeed is a limitation, which we point out in the limitations section of the original manuscript. We recommend exercising caution when selecting specific text-to-image models. It is important to understand the performance characteristics and limitations of each model during the evaluation process.
>
> **[A2] Bias towards INVERT:** We acknowledge the concern regarding the potential bias of the INVERT method toward achieving higher AUC scores in the CoSy evaluation. To give a fair and balanced assessment of explanation quality, we, therefore, introduced the MAD metric as an additional evaluation metric.
> Here, it is also important to note that CLIP-Dissect, another method included in our study, is biased toward CLIP models (also used in Stable Diffusion). By employing multiple evaluation metrics and methods, we strive to offer a comprehensive and unbiased evaluation of performance. For more details on the correlation of AUC and MAD, please refer to our answer [Q1] to R4.
>
> **[A3] Lower-level Concepts:** We also thank R2 for putting further emphasis on the important issue of lower-level concepts. We share the reviewer's interest and plan to extend existing discussions in Section 5.2. The performance of generative models might be a fundamental problem for abstract concepts, as discussed in our limitations. To assess this concern and provide a basis for further discussion, we have also added an additional example of lower-layer concepts in the same style as Figure 5 (see attached PDF, Figure 1). We agree that the generation model faces an increasingly harder task to generate images corresponding to more abstract concepts, which in part contributes to the larger error bars for lower layers. However, we argue that the limitations of text-to-image are still not the fundamental issue even for lower layers as our example shows that explanation methods often fail to provide abstract concepts and present widely different concepts, such that we can still distinguish individual performances. We also performed further experiments regarding low-level concepts and discussed this topic in our response [A3] to R1.

---

### Official Review · Reviewer_1Sdz · 2024-07-13

**Soundness:** 3
**Presentation:** 4
**Contribution:** 3
**Rating:** 7
**Confidence:** 4

**Summary:**

This paper proposes an automatic evaluation for textual explanations of neurons in vision models. The evaluation works by using a text-to-image model to generate images based on the explanation of a neuron. Then, these images are passed through the vision model and that neuron’s activations are recorded. These activations are compared to neuron activations on control images that should have nothing to do with the explanation. A good explanation should yield generated images that produce high neuron activations, as compared to control images. Two metrics are used to measure the difference in neuron activations. The paper validates the evaluation framework by checking that (1) generated images are similar to natural images of class concepts, (2) vision model neuron activations are similar for generated and natural images, and (3) that “ground-truth” explanations (class concept labels) receive high scores while “random” explanations (random class labels) receive low scores. Finally, the authors apply their evaluation to several prominent neuron explanation methods and draw some conclusions about which methods work better and at which layers.

**Strengths:**

- Very important: This paper provides an automatic evaluation for textual explanations of neurons that should provide a reasonably reliable signal for explanation quality. Such an evaluation should serve to guide methods research in the area, and has been sorely needed.
- Very important: The paper tackles an important problem, evaluation of explanation methods in computer vision.
- Important: The paper is very clear, well-organized, and has good illustrations.
- Important: Many small design choices in the paper are very reasonable, like the choice of metrics and prompt selection.
- Important: The paper provides some empirical results with current explanation methods that highlight directions for future research in the area.

**Weaknesses:**

- Very important: The meta-evaluation in this paper suffers a bit on two fronts. First, it is difficult to say it is a meta-evaluation in the sense that we could plug another evaluation procedure into this framework and mesaure how different evaluation results correlate with some *third,* more ground-truth or utility-driven evaluation of explanation methods. Take for example the meta-evaluation of automatic machine-translation metrics. Metrics like BLEU and ROUGE are accepted on the basis of their correlation (or lack thereof) with human judgment of translation quality. What would make the meta-evaluation in this paper more of a meta-evaluation is if it compared with, for instance, evaluations from prior work described in “Prior Methods for Evaluation”, using some third ground-truth/utility-oriented evaluation as a target measure (see e.g. evaluations in https://arxiv.org/pdf/2211.10154 as possible targets). To clarify why I don’t think the results in Sec. 4.3 count as such a ground-truth evaluation, my second point here is that there is a major distribution shift in what we want to measure between typical uses of textual neuron explanations and the validations conducted in Sec. 4. Specifically, Sec. 4 focuses exclusively on evaluations of “output neurons”, i.e. class logits, whereas textual neuron explainers are applied almost exclusively to intermediate hidden neurons in models. This means that even a proper meta-evaluation comparing CoSY against competing explanation methods would be limited by a narrow focus on output neurons and not intermediate neurons. I wonder, could the paper inclue a meta-evaluation experiment utilizing a model that has been trained with intermediate representation supervision, like a kind of concept-bottleneck model, so there could be some ground-truth label for an intermediate neuron (at least as ground-truth as using output neurons, which are learned with direct supervision)?
- Important: I have some other doubts about how the evaluation will work for neurons that represent low-level (perhaps early layer neurons) or abstract features (possibly middle/later layer neurons). For instance, how about low-level edge/shape features, or abstract features like “white objects”. It seems guaranteed that the control images will share many low-level features with the generated images. And it seems very possible that the control images include white objects. These cases seem problematic in that they could lead to low AUC or MAD scores even if the explanation is correct. Can the control images be generated in a conditional way that encourages them to lack the concepts in the proposed explanations?
- Important: Concepts could overlap or be nested in semantic hierarchies, and I am not sure that the evaluation framework could clearly discriminate between good and bad explanations in these settings. For example, we might want to distinguish between “red apples”, “red fruits”, and “red objects” as explanations for a neuron. In order to do this, we need the generated images for each explanation to achieve a sufficient amount of diversity. It would be a problem if when the text-to-image model gets the phrase “red fruits”, it generates mostly red apples (a very salient example of red fruit). So, I would suggest that one meta-evaluation experiment focus on measuring diversity or coverage of the generated images.

**Questions:**

- There has been a shift in the literature from treating neurons as the unit of analysis to feature directions, starting with TCAV (https://arxiv.org/abs/1711.11279) and extending through work like CRAFT (https://arxiv.org/pdf/2211.10154). One example of automatic textual explanation of feature directions is given for LLMs in Sec. 3 of https://arxiv.org/pdf/2309.08600 (as well as https://transformer-circuits.pub/2023/monosemantic-features#global-analysis-interp-auto-acts). I don’t know if any existing methods for vision models could be adapted to focus on latent feature directions rather than neurons, but this would be a ripe direction for future methods.
- Based on the above, the related work section might also point to (1) https://arxiv.org/pdf/2309.08600, (2) https://transformer-circuits.pub/2023/monosemantic-features#global-analysis-interp-auto-acts, and (3) https://arxiv.org/pdf/2211.10154, and possibly point to automatic, model-based evaluations of local explanations like https://arxiv.org/pdf/2312.12747.
- I’m curious what you would think of fitting a logistic classifier to the neuron activations and computing that classifier’s AUC. How would that differ from your non-parametric AUC?

**Limitations:**

I think the limitations discussion is sufficient.

---

> ### Author Rebuttal · Authors · 2024-08-07
>
> We appreciate Reviewer 1 (R1) for their time taken and their great attention to detail shown in the review. We are honored by the positive feedback, that our work was found to be highly relevant and impactful. In the following, we address the reviewer's comments in detail.
>
> **[A1] Meta-Evaluation:**
>
> - **Choice of wording:**  We strongly agree with R1 in their assessment of the term Meta-Evaluation. We apologize that this terminology was misleading. Due to the lack of ground-truth, we should not use the term Meta-Evaluation and therefore replaced it with Sanity Checks.
>
> - **Comparison to prior evaluation:**
> Generally, prior evaluation methods can be divided into two major groups: evaluations based on human studies and evaluations based on assumed ''ground-truth'' labels. Due to the variety of evaluation approaches, a comprehensive comparison in terms of a meta-evaluation is not established. However, we discuss a qualitative comparison to previous methods in response to R4 [A1].
>
> - **Concept-bottleneck for lower-level representations:**
> While we agree with the reviewer's concern regarding lower layer representations and share the interest in concept-bottleneck (CB) based meta-evaluation results, we initially refrained from such experiments. We believe the ground-truth nature of CB concepts and last-layer labels to be often similar. Specifically, mostly during training, intermediate concepts are learned in a supervised manner, often using a similar loss function as the output of the model [1]. Therefore we consider that there is little difference between CB neurons and output neurons.
>
> **[A2] Similarity of generated concepts to control data:**
> We thank R1 for the detailed questions. We acknowledge that the control dataset may indeed contain images visually similar to those in the generated explanations. However, this is not a problem, since the control dataset primarily serves as a collection of random images that represent a diverse array of concepts. While the control dataset might include some images of “white objects”, the majority of images do not include “white objects”. Therefore, for an accurate explanation, the generated images all significantly activate certain neurons, whereas most control images will not. This distinction will be correctly reflected in the AUC and MAD metrics.
>
> Regarding the performance of our methods with low-level neurons, we attribute the low AUC/MAD to the fact that these methods typically output semantically high-level concepts (see Figure 1 in the PDF). This may also be related to the inherent difficulty in describing low-level abstractions in natural language due to their complexity. We will include a broader discussion of this point in the manuscript.
>
> **[A3] Diversity of generated images or concept broadness:**
> We thank reviewer 1 for bringing up the important consideration of the diversity in generated images per concept and the problem of semantically similar concepts. We will include this topic in more detail in the final manuscript. To this end, we will reference our results in Section 5, and Figures 1 (red objects and pomegranate) and 5 (last line), which show that even different concepts with high perceivable image similarity result in separable evaluation results. Moreover, we will discuss that, semantically similar concepts can warrant similar evaluation scores given similar meanings. Thus, differentiation between justified and failed evaluation for semantically similar concepts would require a ground-truth label and even then can be hard to argue. Regarding the diversity of generated images for a single concept, Appendix A.8 and Figure 9 show that the diversity of the generated images does not depend on the chosen prompt and concept, but rather on the temperature value (value of entropy) in the diffusion model. Thus, we see a high similarity of images generated for the same concept.
>
> **[Q1] Feature Directions:** We appreciate the reviewer’s detailed question and thank them. Our evaluation procedure can be applied to extracted concepts or any scalar function within the model, such as linear or nonlinear combinations of neurons, for example, CRAFT. The choice for evaluating the neurons (i.e. canonical basis) and not concepts lies in the fact that the respective publications of most methods explain neurons specifically, with even more limited approaches such as FALCON [2] that can only be applied to certain ''explainable'' neurons. Thus we considered only the canonical basis, to establish a fair comparison. Nonetheless, we are very much interested in extending our work in this direction and plan to address latent feature directions in future work.
>
> **[Q2] Related Work:**  We thank the reviewer for pointing out these important works, which we will add to the related work section.
>
> **[Q3] Fitting a logistic classifier to the neuron activations:** We appreciate the reviewer’s suggestion and the opportunity to address this point. However, we kindly request further clarification regarding the proposal. Specifically, could the reviewer provide more details on what is meant by applying a logistic classifier prior to computing the AUC?
>
> [1] Koh, Pang Wei, et al. ''Concept bottleneck models.'' ICML, (2020).
>
> [2] Kalibhat, Neha, et al. ''Identifying interpretable subspaces in image representations.'' ICML, 2023.

---

> > ### Comment · Reviewer_1Sdz · 2024-08-10
> > **Response to rebuttal**
> >
> > Thanks for the thorough response! Some comments below:
> >
> > >Choice of wording:…Sanity Checks
> >
> > Thanks! Sanity Checks sounds good to me.
> >
> > >Comparison to prior evaluation:
> >
> > Thanks, I’m pretty sympathetic to the point that actually comparing against other evals fairly is difficult here. It seems acceptable, though not totally ideal, to judge how reasonable each evaluation framework seems on the merits.
> >
> > >Concept-bottleneck for lower-level representations: ...We believe the ground-truth nature of CB concepts and last-layer labels to be often similar. Specifically, mostly during training, intermediate concepts are learned in a supervised manner, often using a similar loss function as the output of the model [1]. Therefore we consider that there is little difference between CB neurons and output neurons.
> >
> > Well, a concept could be “fur” or “leg”. I think these concepts are much more low-level than the classes in ImageNet, which includes many specific animals.
> >
> > I agree that the loss functions may be similar.
> >
> > Overall, I think there is still some important distribution shift between intermediate (CB) concepts and class/label concepts.
> >
> > > [A2] Similarity of generated concepts to control data:
> >
> > Ok I agree with the claim about “white objects”. The control data should have few white objects, while the hypothesis data could have many, and MAD should reflect this.
> >
> > But I am still concerned about cases with other low level features, like edge detectors. I agree that with the claim about “inherent difficulty in describing low-level abstractions in natural language due to their complexity.“ It may be hard to describe an edge detector in words. But it would be nice to have evals that reflect this and penalize methods that fail to do this, if that is what a feature truly represents! The concern right now is that both the generated and control images will have a bunch of edges in them. So I think an even better eval would try to *remove* the the hypothesis feature from the control images.
> >
> > >[A3] Diversity of generated images or concept broadness…Appendix A.8 and Figure 9
> >
> > I’m not sure I see A.8 and Fig 9. I see A.6 and Fig 8? Either way, based on A.6 and the rebuttal, I can accept that the generated images seem diverse enough for purposes of rank ordering explanations by their quality. It could be nice to show that the rank ordering of methods is similar across different diffusion model sampling temperatures (as well as other details of the diffusion model, of course), but I’m satisified with the results here.
> >
> > >Thus we considered only the canonical basis, to establish a fair comparison.
> >
> > Makes sense!
> >
> > >[Q3] Fitting a logistic classifier to the neuron activations:
> >
> > Right, so you could take the activations for the control images and activations for the generated images and try to classify them as control vs. generated with a logistic regression. So it’s a regression with one feature, which is the neuron activation. Then you can compute an AUROC for this logistic model.
> >
> > Contrast this with the AUC derived from a pairwise comparison of all possible (a,b) activation pairs across the two groups, which is non-parametric.
> >
> > I actually don’t know what the difference would be. Just an idea.
> >
> > ---
> >
> > Based on the above discussion, I continue to be happy with the paper and maintain my score of 7.

---

> > > ### Author Response · Authors · 2024-08-14
> > >
> > > We appreciate your detailed comment and apologize for the confusion regarding our references in [A3]. You are correct; we intended to refer to Appendix A.6 and Figure 8 in the original manuscript.
> > >
> > > Thank you for clarifying the approach of fitting a logistic classifier to the neuron activations. Comparing AUROC derived from logistic regression with the non-parametric AUC is an interesting idea. We appreciate your suggestion and will consider incorporating this approach in our future analyses.
> > >
> > > Thank you again for your thoughtful feedback and for maintaining your score.

---

### Author Rebuttal · Authors · 2024-08-07

First, we would like to deeply thank all the reviewers for the time they spent reviewing our manuscript. We express our gratitude for the valuable comments and advice. We are honored by their detailed feedback and are strongly encouraged by the generally positive feedback.

We are particularly grateful that all reviewers found our work to be sound and well-presented. We also appreciate reviewers R1, R2, and R4 for recognizing the significance of our contributions and emphasizing the importance of our paper in addressing the critical issue of automatic evaluation for textual explanations of neurons. As R1 highlighted, ''[s]uch an evaluation should serve to guide methods research in the area, and has been sorely needed.'' Additionally, R2 noted that this work is ''critical for the advancement of explainable AI,'' while R4 praised our work as ''the first to propose a unified, intuitive, and valid way of evaluating neuron annotation works.''

Furthermore, we appreciate that reviewers R3 and R2 acknowledged the CoSy framework's architecture-agnostic design, which makes it ''broadly applicable to various models'' (R3). This flexibility enhances the framework's relevance and applicability in the field.

Our empirical results not only present the current state of explanation methods but as R1 noted, it also ''highlight directions for future research,''. Moreover, R4 found our findings ''very interesting and important, which raises a concern in the interpretability field,'' and appreciated that our evaluation framework is ''simple, something which all people and researchers will appreciate.''

Inspired by their helpful comments, we have incorporated the following main changes into the revision:

- We discuss prior evaluation methods in more depth (R1-A1, R3-A2, R4-A1).
- Provide a more detailed explanation of the metrics used, such as AUC, MAD, and image similarity (R3-A3, R3-Q1, R4-A4).
- We have added a normalization term to the MAD metric, increasing the interpretability of the resulting score. Please note that all qualitative results remain unchanged, and there are no changes in the performance rankings of explanation methods. (Equation 1, PDF)
- Use more clear and coherent terminology, including renaming the meta-evaluation section for clarity (R1-A1, R4-A8).
- Include a greater number of concepts in the sanity checks (R3-Q2).
- Expand the benchmark comparison to include additional models (R4-A2).

Below each review, we will address the specific questions and concerns separately. The PDF file containing updated figures and results is attached.

We thank all the reviewers again for their time, effort, and constructive feedback!

---

### Decision · Program_Chairs · 2024-09-25

**Decision:**

Accept (poster)

**Comment:**

This paper proposes a framework to evaluate the text explanations of neurons, Concept Synthesis (CoSy). Given textual explanations, CoSy uses a generative model conditioned on textual input to create data points representing the textual explanation. The neuron’s response towards these data points are compared with the response to control data points to provide a quality estimate of the given explanation. This is the first general-purpose, quantitative evaluation framework that enables the evaluation of explanations for CV models. With extensive meta-evaluation experiments, this work analyze the choice of generative models, prompts, and explanation methods.

The reviewers agree on the positive ratings of this paper, and the rebuttals addressed most concerns. That said, the reviewers mentioned some aspects of improvement, e.g., about the empirical rigor of the evaluation procedure and the coverage of the concepts. There can be further inquiry into the mechanisms (and further, the developments) of the models. But this paper itself already marks a solid, significant and valuable contribution.